# A Unified Approach to Count-Based Weakly-Supervised Learning

**Vinay Shukla**
Department of Computer Science
University of California, Los Angeles
vshukla@g.ucla.edu

**Zhe Zeng***
Department of Computer Science
University of California, Los Angeles
zhezeng@cs.ucla.edu

**Kareem Ahmed***
Department of Computer Science
University of California, Los Angeles
ahmedk@cs.ucla.edu

**Guy Van den Broeck**
Department of Computer Science
University of California, Los Angeles
guyvdb@cs.ucla.edu

## Abstract

High-quality labels are often very scarce, whereas unlabeled data with inferred weak labels occurs more naturally. In many cases, these weak labels dictate the frequency of each respective class over a set of instances. In this paper, we develop a unified approach to learning from such weakly-labeled data, which we call *count-based weakly-supervised learning*. At the heart of our approach is the ability to compute the probability of exactly $k$ out of $n$ outputs being set to true. This computation is differentiable, exact, and efficient. Building upon the previous computation, we derive a *count loss* penalizing the model for deviations in its distribution from an arithmetic constraint defined over label counts. We evaluate our approach on three common weakly-supervised learning paradigms and observe that our proposed approach achieves state-of-the-art or highly competitive results across all three of the paradigms.

## 1 Introduction

Weakly supervised learning [56] enables a model to learn from data with restricted, partial or inaccurate labels, often known as *weakly-labeled data*. Weakly supervised learning fulfills a need arising in many real-world settings that are subject to privacy or budget constraints, such as privacy sensitive data [45], medical image analysis [12], clinical practice [39], personalized advertisement [9] and knowledge base completion [21, 59], to name a few. In some settings, *instance-level labels* are unavailable. Instead, instances are grouped into *bags* with corresponding *bag-level labels* that are a function of the instance labels, e.g., the proportion of positive labels in a bag. A key insight that we bring forth is that such weak supervision can very often be construed as *enforcing constraints on label counts of data*.

More concretely, we consider three prominent weakly supervised learning paradigms. The first paradigm is known as *learning from label proportions* [38]. Here the weak supervision consists in the *proportion* of positive labels in a given bag, which can be interpreted as *the count of positive instances* in such a bag. The second paradigm, whose supervision is strictly weaker than the former, is *multiple instance learning* [35, 17]. Here the bag labels only indicate the *existence* of at least one positive instance in a bag, which can be recast as to whether *the count of positive instances* is greater than zero. The third paradigm, *learning from positive and unlabeled data* [16, 31], grants access to

---

*Equal contribution.

37th Conference on Neural Information Processing Systems (NeurIPS 2023).

| $\boldsymbol{x}$ | $y$ |
|---|---|
| (digit) | 0 |
| (digit) | 0 |
| (digit) | 1 |
| (digit) | 1 |

(a) Classical

| $\{\boldsymbol{x}_i\}_{i=1}^k$ | $\tilde{y} = \sum y_i/k$ |
|---|---|
| (digits) | 0 |
| (digits) | 1/3 |
| (digits) | 3/5 |

(b) LLP

| $\{\boldsymbol{x}_i\}_{i=1}^k$ | $\tilde{y} = \max\{y_i\}$ |
|---|---|
| (digits) | 0 |
| (digits) | 1 |
| (digits) | 1 |

(c) MIL

| $\boldsymbol{x}$ | $\tilde{y}$ |
|---|---|
| (digit) | ? |
| (digit) | 1 |
| (digit) | ? |
| (digit) | ? |

(d) PU Learning

Table 1: A comparison of the tasks considered in the three weakly supervised settings, LLP (cf. Section 2.1), MIL (cf. Section 2.2) and PU learning (cf. Section 2.3), against the classical fully supervised setting for binary classification, using digits from the MNIST dataset.

the ground truth labels for a subset of *only the positive instances*, providing only a class prior for what remains. We can recast the class prior as *a distribution of the count of positive labels*.

Leveraging the view of weak supervision as a constraint on label counts, we utilize a simple, efficient and probabilistically sound approach to weakly-supervised learning. More precisely, we train a neural network to make instance-level predictions that conform to the desired label counts. To this end, we propose a *differentiable count loss* that characterizes how close the network's distribution comes to the label counts; a loss which is surprisingly tractable. Compared to prior methods, this approach does not approximate probabilities but computes them *exactly*. Our empirical evaluation demonstrates that our proposed count loss significantly boosts the classification performance on all three aforementioned settings.

# 2 Problem Formulations

In this section, we formally introduce the aforementioned weakly supervised learning paradigms. For notation, let $\mathcal{X} \in \mathbb{R}^d$ be the input feature space over $d$ features and $\mathcal{Y} = \{0, 1\}$ be a binary label space. We write $\boldsymbol{x} \in \mathcal{X}$ and $y \in \mathcal{Y}$ for the input and output random variables respectively. Recall that in fully-supervised binary classification, it is assumed that each feature and label pair $(\boldsymbol{x}, y) \in \mathcal{X} \times \mathcal{Y}$ is sampled independently from a joint distribution $p(\boldsymbol{x}, y)$. A classifier $f$ is learned to minimize the risk $R(f) = \mathbb{E}_{(\boldsymbol{x}, y) \sim p}[\ell(f(\boldsymbol{x}), y)]$ where $\ell : [0, 1] \times \mathcal{Y} \to \mathbb{R}_{\geq 0}$ is the cross entropy loss function. Typically, the true distribution $p(\boldsymbol{x}, y)$ is implicit and cannot be observed. Therefore, a set of $n$ training samples, $\mathcal{D} = \{(\boldsymbol{x}_i, y_i)\}_{i=1}^n$, is used and the empirical risk, $\hat{R}(f) = \frac{1}{n}\sum_{i=1}^n \ell(f(\boldsymbol{x}_i), y_i)$, is minimized in practice. In the count-based weakly supervised learning settings, the supervision is given at a bag level instead of an instance level. We formally introduce these settings as below.

## 2.1 Learning from Label Proportions

*Learning from label proportions (LLP)* [38] assumes that each instance in the training set is assigned to bags and only the proportion of positive instances in each bag is known. One example is in light of the coronavirus pandemic, where infection rates were typically reported based on geographical boundaries such as states and counties. Each boundary can be treated as a bag with the infection rate as the proportion annotation.

The goal of LLP is to learn an instance-level classifier $f : \mathcal{X} \to [0, 1]$ even though it is trained on bag-level labeled data. Formally, the training dataset consists of $m$ bags, denoted by $\mathcal{D} = \{(B_i, \tilde{y}_i)\}_{i=1}^m$ where each bag $B_i = \{\boldsymbol{x}_j\}_{j=1}^k$ consist of $k$ instances and this $k$ could vary among different bags. The bag proportions are defined as $\tilde{y}_i = \sum_{j=1}^k y_j/k$ with $y_j$ being the instance label that cannot be accessed and only $\tilde{y}_i$ is available during training. An example is shown in Figure 1b. We do not assume that the bags are non-overlapping while some existing work suffers from this limitation including Scott and Zhang [40].

Table 2: A summary of the labels and objective functions for all the settings considered in the paper.

| TASK | LABEL | LABEL LEVEL | OBJECTIVE |
|---|---|---|---|
| Classical Fully Supervised | Binary $y$ | Instance Level | $-y \log p(y) - (1-y) \log(1 - p(y))$ |
| Learning from Label Proportion | Continuous $\tilde{y} = \sum_i y_i / k$ | Bag Level | $-\log p(\sum \hat{y}_i = k\tilde{y})$ |
| Multiple Instance Learning | Binary $\tilde{y} = \max\{y_i\}$ | Bag Level | $-\tilde{y} \log p(\sum \hat{y}_i \geq 1) - (1 - \tilde{y}) \log p(\sum_i \hat{y}_i = 0)$ |
| Learning from Positive and Unlabeled Data | Binary $\tilde{y}$ | Instance Level | 1) $\mathbb{D}_{KL}(\text{Bin}(k, \beta) \parallel p(\sum_i \hat{y}_i))$ 
 2) $-\log p(\sum \hat{y}_i = k\beta)$ |

## 2.2 Multiple Instance Learning

*Multiple instance learning (MIL)* [35, 17] refers to the scenario where the training dataset consists of bags of instances, and labels are provided at bag level. However, in MIL, the bag label is a single binary label indicating whether there is a positive instance in the bag or not as opposed to a bag proportion defined in LLP. A real-world application of MIL lies in the field of drug activity [17]. We can observe the effects of a group of conformations but not for any specific molecule, motivating a MIL setting. Formally, in MIL, the training dataset consists of $m$ bags, denoted by $\mathcal{D} = \{(B_i, \tilde{y}_i)\}_{i=1}^m$, with a bag consisting of $k$ instances, i.e., $B_i = \{\boldsymbol{x}_j\}_{j=1}^k$. The size $k$ can vary among different bags. For each instance $\boldsymbol{x}_j$, there exists an instance-level label $y_j$ which is not accessible. The bag-level label is defined as $\tilde{y}_i = \max_j\{y_j\}$. An example is shown in Figure 1c.

The main goal of MIL is to learn a model that predicts a bag label while a more challenging goal is to learn an instance-level predictor that is able to discover positive instances in a bag. In this work, we aim to tackle both by training an instance-level classifier whose predictions can be combined into a bag-level prediction as the last step.

## 2.3 Learning from Positive and Unlabeled Data

*Learning from positive and unlabeled data* or *PU learning* [16, 31] refers to the setting where the training dataset consists of only positive instances and unlabeled data, and the unlabeled data can contain both positive and negative instances. A motivation of PU learning is persistence in the case of shifts to the negative-class distribution [37], for example, a spam filter. An attacker may alter the properties of a spam email, making a traditional classifier require a new negative dataset [37]. We note that taking a new unlabeled sample would be more efficient, motivating PU learning. Formally, in PU learning, the training dataset $\mathcal{D} = \mathcal{D}_p \cup \mathcal{D}_u$ where $\mathcal{D}_p = \{(\boldsymbol{x}_i, \tilde{y}_i = 1)\}_{i=1}^{n_p}$ is the set of positive instances with $\boldsymbol{x}_i$ from $p(\boldsymbol{x} \mid y = 1)$ and $\tilde{y}$ denoting whether the instance is labeled, and $\mathcal{D}_u = \{(\boldsymbol{x}_i, \tilde{y}_i = 0)\}_{i=1}^{n_u}$ the unlabeled set with $\boldsymbol{x}_i$ from

$$p_u(\boldsymbol{x}) = \beta\, p(\boldsymbol{x} \mid y = 1) + (1 - \beta)\, p(\boldsymbol{x} \mid y = 0), \tag{1}$$

where the mixture proportion $\beta := p(y = 1 \mid \tilde{y} = 0)$ is the fraction of positive instances among the unlabeled population. Although the instance label $y$ is not accessible, its information can be inferred from the binary selection label $\tilde{y}$: if the selection label $\tilde{y} = 1$, it belongs to the positively labeled set, i.e., $p(y = 1 \mid \tilde{y} = 1) = 1$; otherwise, the instance $\boldsymbol{x}$ can be either positive or negative. An example of such a dataset is shown in Figure 1d.

The goal of PU learning is to train an instance-level classifier. However, it is not straightforward to learn from PU data and it is necessary to make assumptions to enable learning with positive and unlabeled data [9]. In this work, we make a commonly-used assumption for PU learning, *selected completely at random (SCAR)*, which lies at the basis of many PU learning methods.

**Definition 2.1** (SCAR). Labeled instances are selected completely at random, independent from input features, from the positive distribution $p(\boldsymbol{x} \mid y = 1)$, that is, $p(\tilde{y} = 1 \mid \boldsymbol{x}, y = 1) = p(\tilde{y} = 1 \mid y = 1)$.

## 3 A Unified Approach: Count Loss

In this section, we derive objectives for the three weakly supervised settings, LLP, MIL, and PU learning, from first principles. Our proposed objectives bridge between neural outputs, which can be observed as counts, and arithmetic constraints derived from the weakly supervised labels. The idea is to capture how close the classifier is to satisfying the arithmetic constraints on its outputs.

---

**Algorithm 1** Count Probability $p(\sum_{i=1}^{k} \hat{y}_i = s)$

---

**Input:** A set of $k$ log probabilities $\{t_i\}_{i=1}^{k}$ with $t_i := \log p(\hat{y}_i = 1)$, the number of instances $k$, and a label sum $s$

**Output:** log probabilities $\log p(\sum_{i=1}^{k} \hat{y}_i = s)$ or a set of log probability $\{\log p(\sum_{i=1}^{k} \hat{y}_i = s)\}_{s=0}^{k}$

    // $A[i, m] = \log p(\sum_{j=1}^{i} y_j = m) \; \forall i, m$

    Initialize an array $A$ to be $-\mathsf{Inf}$ everywhere

    $A[0, 0] = 0$    // $p(\sum_{j=1}^{0} y_j = 0) = 1$

    Compute $t_i' \leftarrow \texttt{log1mexp}(t_i)$ // $\log p(y_i = 0)$

    **for** $i = 1$ **to** $k$ **do**

        **for** $m = 0$ **to** $s$ **do**

            $a_+ = A[i - 1, m - 1] + t_i$

            $a_- = A[i - 1, m] + t_i'$

            $A[i, m] = \texttt{logsumexp}(a_+, a_-)$

    **return** $A[k, s]$ or $A[k, :]$

---

They can be easily integrated with deep learning models, and allow them to be trained end-to-end. For the three objectives, we show that they share the same computational building block: given $k$ instances $\{\boldsymbol{x}_i\}_{i=1}^{k}$ and an instance-level classifier $f$ that predicts $p(\hat{y}_i \mid \boldsymbol{x}_i)$ with $\hat{y}$ denoting the prediction variable, the problem of inferring the probability of the constraint on counts $\sum_{i=1}^{k} \hat{y}_i = s$ is to compute the count probability defined below:

$$p(\sum_{i=1}^{k} \hat{y}_i = s \mid \{\boldsymbol{x}_i\}_{i=1}^{k}) := \sum_{\hat{\boldsymbol{y}} \in \mathcal{Y}^k} [\![\sum_{i=1}^{k} \hat{y}_i = s]\!] \prod_{i=1}^{k} p(\hat{y}_i \mid \boldsymbol{x}_i)$$

where $[\![\cdot]\!]$ denotes the indicator function and $\hat{\boldsymbol{y}}$ denotes the vector $(\hat{y}_1, \cdots, \hat{y}_k)$. For succinctness, we omit the dependency on the input and simply write the count probability as $p(\sum_{i=1}^{k} \hat{y}_i = s)$. Next, we show how the objectives derived from first principles can be solved by using the count probability as an oracle. We summarize all proposed objectives in Table 2. Later, we will show how this seemingly intractable count probability can be efficiently computed by our proposed algorithm.

**LLP setting.** Given a bag $B = \{\boldsymbol{x}_i\}_{i=1}^{k}$ of size $k$ and its weakly supervised label $\tilde{y}$, by definition, it can be inferred that the number of positive instances (count) in the bag is $k\tilde{y}$. Our objective is to minimize the negative log probability $-\log p(\sum_i \hat{y}_i = k\tilde{y})$. Notice that when each bag consists of only one instance, that is, when the bag-level supervisions are reduced to instance-level ones, this objective is exactly cross-entropy loss. We further show that our method is risk-consistent, that is, the optimal classifier under our proposed loss provides predictions consistent with the underlying risk as in the supervised learning setting. Details of the risk analysis can be found in Appendix A.

**MIL setting.** Given a bag $B = \{\boldsymbol{x}_i\}_{i=1}^{k}$ of size $k$ and a single binary label $\tilde{y}$ as its weakly supervised label, we propose a cross-entropy loss as below

$$\ell(B, \tilde{y}) = -\tilde{y} \log p(\sum \hat{y}_i \geq 1) - (1 - \tilde{y}) \log p(\sum \hat{y}_i = 0).$$

Notice that in the above loss, the probability term $p(\sum \hat{y}_i = 0)$ is accessible to the oracle for computing count probability, and the other probability term $p(\sum \hat{y}_i \geq 1)$ can simply be obtained from $1 - p(\sum \hat{y}_i = 0)$, i.e., the same call to the oracle since all prediction variables $\hat{y}_i$ are binary.

**PU Learning setting.** Recall that for the unlabeled data $\mathcal{D}_u$ in the training dataset, an unlabeled instance $\boldsymbol{x}_i$ is drawn from a mixture distribution as shown in Equation 1 parameterized by a mixture proportion $\beta = p(y = 1 \mid \tilde{y} = 0)$. Under the SCAR assumption, even though only a class prior is given, we show that the mixture proportion can be estimated from the dataset.

**Proposition 3.1.** *With SCAR assumption and a class prior* $\alpha := p(y = 1)$, *the mixture proportion* $\beta := p(y = 1 \mid \tilde{y} = 0)$ *can be estimated from dataset* $\mathcal{D}$.

*Proof.* First, the label frequency $p(\tilde{y} = 1 \mid y = 1)$ denoted by $c$ can be obtained by

$$c = \frac{p(\tilde{y} = 1, y = 1)}{p(y = 1)} = \frac{p(\tilde{y} = 1)}{p(y = 1)} \;\; \textit{(by the definition of PU learning)}.$$

Figure 1: An example of how to compute the count probability in a dynamic programming manner. Assume that an instance-level classifier predicts three instances to have $p(y_1 = 1) = 0.1$, $p(y_2 = 1) = 0.2$, and $p(y_3 = 1) = 0.3$ respectively. The algorithm starts from the top-left cell and propagates the results down right. A cell has its probability $p(\sum_{j=0}^{i} y_j = s)$ computed by inputs from $p(\sum_{j=0}^{i-1} y_j = s)$ weighted by $p(y_i = 0)$, and $p(\sum_{j=0}^{i-1} y_j = s - 1)$ weighted by $p(y_i = 1)$ respectively, as indicated by the arrows.

that is, $c = p(\tilde{y} = 1)/\alpha$. Notice that $p(\tilde{y} = 1)$ can be estimated from the dataset $\mathcal{D}$ by counting the proportion of the labeled instances. Thus, we can estimate the mixture proportion as below,

$$\beta = \frac{p(\tilde{y} = 0 \mid y = 1)p(y = 1)}{p(\tilde{y} = 0)} = \frac{(1 - p(\tilde{y} = 1 \mid y = 1))p(y = 1)}{1 - p(\tilde{y} = 1)} = \frac{(1 - c)\alpha}{1 - \alpha c}.$$

$\square$

The probabilistic semantic of the mixture proportion is that if we randomly draw an instance $x_i$ from the unlabeled population, the probability that the true label $y_i$ is positive would be $\beta$. Further, if we randomly draw $k$ instances, the distribution of the summation of the true labels $\sum_{i=1}^{k} y_i$ conforms to a binomial distribution $\texttt{Bin}(k, \beta)$ parameterized by the mixture proportion $\beta$, i.e.,

$$p(\sum_{i=1}^{k} y_i = s) = \binom{k}{s} \beta^s (1 - \beta)^{k-s}. \tag{2}$$

Based on this observation, we propose an objective to minimize the KL divergence between the distribution of predicted label sum and the binomial distribution parameterized by the mixture proportion for a random subset drawn from the unlabeled population, that is,

$$\mathbb{D}_{KL}\left(\texttt{Bin}(k, \beta) \parallel p(\sum_{i=1}^{k} \hat{y}_i)\right) = \sum_{s=0}^{k} \texttt{Bin}(s; k, \beta) \log \frac{\texttt{Bin}(s; k, \beta)}{p(\sum_{i=1}^{k} \hat{y}_i = s)}$$

where $\texttt{Bin}(s; k, \beta)$ denotes the probability mass function of the binomial distribution $\texttt{Bin}(k, \beta)$. Again, the KL divergence can be obtained by $k + 1$ calls to the oracle for computing count probability $p(\sum_{i=1}^{k} \hat{y}_i = s)$. The KL divergence is further combined with a cross entropy defined over labeled data $\mathcal{D}_p$ as in the classical binary classification training as the overall objective.

As an alternative, we propose an objective for the unlabeled data that requires fewer calls to the oracle: instead of matching the distribution of the predicted label sum with the binomial distribution, this objective matches only the expectations of the two distributions, that is, to maximize $p(\sum_{i=1}^{k} \hat{y}_i = k\beta)$ where $k\beta$ is the expectation of the binomial distribution $\texttt{Bin}(k, \beta)$. We present empirical evaluations of both proposed objectives in the experimental section.

## 4 Tractable Computation of Count Probability

In the previous section, we show how the count probability $p(\sum_{i=1}^{k} \hat{y}_i = s)$ serves as a computational building block for the objectives derived from first principles for the three weakly supervised learning settings. With a closer look at the count probability, we can see that given a set of instances, the classifier predicts an instance-level probability for each and it requires further manipulation to obtain

count information; actually, the number of joint labelings for the set can be exponential in the number of instances. Intractable as it seems, we show that it is indeed possible to derive a tractable computation for the count probability based on a result from Ahmed et al. [6].

**Proposition 4.1.** *The count probability $p(\sum_{i=1}^{k} \hat{y}_i = s)$ of sampling $k$ prediction variables that sums to $s$ from an unconstrained distribution $p(\boldsymbol{y}) = \prod_{i=1}^{k} p(\hat{y}_i)$ can be computed exactly in time $\mathcal{O}(ks)$. Moreover, the set $\{p(\sum_{i=1}^{k} \hat{y}_i = s)\}_{s=0}^{k}$ can also be computed in time $\mathcal{O}(k^2)$.*

The above proposition can be proved in a constructive way where we show that the count probability $p(\sum_{i=1}^{k} \hat{y}_i = s)$ can be computed in a dynamic programming manner. We provide an illustrative example of this computation in Figure 1. In practice, we implement this computation in log space for numeric stability which we summarized as Algorithm 1, where function `log1mexp` provides a numerically stable way to compute $\text{log1mexp}(x) = \log(1 - \exp(x))$ and function `logsumexp` a numerically stable way to compute $\text{logsumexp}(x, y) = \log(\exp(x) + \exp(y))$. Notice that since we show it is tractable to compute the set $\{p(\sum_{i=1}^{k} \hat{y}_i = s)\}_{s=0}^{k}$, for any two given label sum $s_1$ and $s_2$, a count probability $p(s_1 \leq \sum_i \hat{y}_i \leq s_2)$ where the count lies in an interval, can also be exactly and tractably computed. This implies that our tractable computation of count probabilities can potentially be leveraged by other count-based applications besides the three weakly supervised learning settings in the last section.

## 5 Related Work

**Weakly Supervised Learning.** Besides settings explored in our work there are many other weakly-supervised settings. One of which is semi-supervised learning, a close relative to PU Learning with the difference being that labeled samples can be both positive and negative [57, 58]. Another is label noise learning, which occurs when our instances are mislabeled. Two common variations involve whether noise is independent or dependent on the instance [20, 42]. A third setting is partial label learning, where each instance is provided a set of labels of which exactly one is true [14]. An extension of this is partial multi-label learning, where among a set of labels, a subset is true [46].

**Unified Approaches.** There exists some literature in regards to "general" approaches for weakly supervised learning. One example being the method proposed in Hüllermeier [23], which provides a procedure that minimizes the empirical risk on "fuzzy" sets of data. The paper also establishes guarantees for model identification and instance-level recognition. Co-Training and Self-Training are also examples of similar techniques that are applicable to a wide variety of weakly supervised settings [11, 49]. Self-training involves progressively incorporating more unlabeled data via our model's prediction (with pseudo-label) and then training a model on more data as an iterative algorithm [25]. Co-Training leverages two models that have different "views" of the data and iteratively augment each other's training set with samples they deem as "well-classified". They are traditionally applied to semi-supervised learning but can extend to multiple instance learning settings [33, 47, 32].

**LLP.** Quadrianto et al. [38] first introduced an exponential family based approach that used an estimation of mean for each class. Others seek to minimize "empirical proportion risk" or EPR as in Yu et al. [50], which is centered around creating an instance-level classifier that is able to reproduce the label proportions of each bag. As mentioned previously, more recent methods use bag posterior approximation and neural-based approaches [8, 43]. One such method is Proportion Loss (PL) [43], which we contrast to our approach. This is computed by binary cross entropy between the averaged instance-level probabilities and ground-truth bag proportion.

**MIL.** MIL finds its earlier approaches with SVMs, which have been used quite prolifically and still remain one of the most common baselines. We start with MI-SVM/mi-SVM [7] which are examples of transductive SVMs [13] that seek a stable instance classification through repeated retraining iterations. MI-SVM is an example of an instance space method [13], which identifies methods that classify instances as a preliminary step in the problem. This is in contrast to bag-space or embedded-space methods that omit the instance classification step. Furthermore, Wang et al. [44] remains one of the hallmarks of the use of neural networks for Multi-Instance Learning. Ilse et al. [24], utilize a similar approach but with attention-based mechanisms.

**PU learning.** Bekker and Davis [9] groups PU Learning paradigms into three main classes: two step, biased, and class prior incorporation. Biased learning techniques train a classifier on the entire dataset

Table 3: LLP results across different bag sizes. We report the mean and standard deviation of the test AUC over 5 seeds for each setting. The highest metric for each setting is shown in **boldface**.

| Dataset | Dist | Method | 8 | 32 | 128 | 512 |
|---------|------|--------|---|----|-----|-----|
| Adult | $[0, \frac{1}{2}]$ | PL | $0.8889 \pm 0.0024$ | $0.8782 \pm 0.0036$ | $\mathbf{0.8743 \pm 0.0039}$ | $0.8678 \pm 0.0085$ |
| Adult | $[0, \frac{1}{2}]$ | LMMCM | $0.8728 \pm 0.0019$ | $0.8693 \pm 0.0047$ | $0.8669 \pm 0.0041$ | $0.8674 \pm 0.0040$ |
| Adult | $[0, \frac{1}{2}]$ | CL (Ours) | $\mathbf{0.8984 \pm 0.0013}$ | $\mathbf{0.8848 \pm 0.0041}$ | $\mathbf{0.8743 \pm 0.0052}$ | $\mathbf{0.8703 \pm 0.0070}$ |
| Adult | $[\frac{1}{2}, 1]$ | PL | $0.8781 \pm 0.0038$ | $0.8731 \pm 0.0035$ | $\mathbf{0.8699 \pm 0.0057}$ | $0.8556 \pm 0.0180$ |
| Adult | $[\frac{1}{2}, 1]$ | LMMCM | $0.8584 \pm 0.0164$ | $0.8644 \pm 0.0052$ | $0.8601 \pm 0.0045$ | $0.8500 \pm 0.0186$ |
| Adult | $[\frac{1}{2}, 1]$ | CL (Ours) | $\mathbf{0.8854 \pm 0.0022}$ | $\mathbf{0.8738 \pm 0.0039}$ | $0.8675 \pm 0.0043$ | $\mathbf{0.8607 \pm 0.0056}$ |
| Adult | $[0, 1]$ | PL | $0.8884 \pm 0.0030$ | $0.8884 \pm 0.0008$ | $\mathbf{0.8879 \pm 0.0025}$ | $\mathbf{0.8828 \pm 0.0051}$ |
| Adult | $[0, 1]$ | LMMCM | $0.8831 \pm 0.0026$ | $0.8819 \pm 0.0006$ | $0.8821 \pm 0.0017$ | $0.8786 \pm 0.0052$ |
| Adult | $[0, 1]$ | CL (Ours) | $\mathbf{0.8985 \pm 0.0010}$ | $\mathbf{0.8891 \pm 0.0013}$ | $0.8871 \pm 0.0021$ | $0.8790 \pm 0.0056$ |
| Magic | $[0, \frac{1}{2}]$ | PL | $0.8900 \pm 0.0095$ | $0.8510 \pm 0.0032$ | $0.8405 \pm 0.0110$ | $0.8332 \pm 0.0149$ |
| Magic | $[0, \frac{1}{2}]$ | LMMCM | $0.8918 \pm 0.0077$ | $0.8799 \pm 0.0113$ | $0.8753 \pm 0.0157$ | $0.8734 \pm 0.0092$ |
| Magic | $[0, \frac{1}{2}]$ | CL (Ours) | $\mathbf{0.9088 \pm 0.0056}$ | $\mathbf{0.8830 \pm 0.0097}$ | $\mathbf{0.8926 \pm 0.0049}$ | $\mathbf{0.8864 \pm 0.0107}$ |
| Magic | $[\frac{1}{2}, 1]$ | PL | $0.9066 \pm 0.0016$ | $0.8818 \pm 0.0108$ | $0.8769 \pm 0.0101$ | $0.8429 \pm 0.0443$ |
| Magic | $[\frac{1}{2}, 1]$ | LMMCM | $0.8911 \pm 0.0083$ | $0.8790 \pm 0.0091$ | $0.8684 \pm 0.0046$ | $0.8567 \pm 0.0292$ |
| Magic | $[\frac{1}{2}, 1]$ | CL (Ours) | $\mathbf{0.9105 \pm 0.0020}$ | $\mathbf{0.8980 \pm 0.0059}$ | $\mathbf{0.8851 \pm 0.0255}$ | $\mathbf{0.8816 \pm 0.0083}$ |
| Magic | $[0, 1]$ | PL | $0.9039 \pm 0.0029$ | $0.8870 \pm 0.0037$ | $0.9002 \pm 0.0092$ | $0.8807 \pm 0.0200$ |
| Magic | $[0, 1]$ | LMMCM | $0.9070 \pm 0.0026$ | $0.9048 \pm 0.0058$ | $0.9113 \pm 0.0058$ | $0.8934 \pm 0.0097$ |
| Magic | $[0, 1]$ | CL (Ours) | $\mathbf{0.9173 \pm 0.0018}$ | $\mathbf{0.9102 \pm 0.0057}$ | $\mathbf{0.9146 \pm 0.0051}$ | $\mathbf{0.9088 \pm 0.0039}$ |

with the understanding that negative samples are subject to noise [9]. We will focus on a subset of biased learning techniques (Risk Estimators) as they are considered state-of-the-art and relevant to us as baselines. The Unbiased Risk Estimator (uPU) provides an alternative to the inefficiencies in manually biasing unlabeled data [18, 37]. Later, Non-negative Risk Estimator (nnPU) [26] accounted for weaknesses in the unbiased risk estimator such as overfitting.

**Count Loss.** To our knowledge, viewing the computation of the "bag posterior" as *probabilistic* is new. However, the prior approaches do this implicitly. Many approaches have tried to approximate the "bag posterior" by averaging the instance-level probabilities in a bag [8, 43]. In MIL settings, among instance-level approaches, the MIL-pooling is an implicit "bag posterior" computation. These include mean, max, and log-sum-exp pooling to approximate the likelihood that a bag has at least one positive instance [44]. But again, these are all approximations of what our computation does *exactly*. In PU Learning, to our best knowledge, the view of unlabeled data as a bag annotated with the mixture proportion is new.

**Neuro-Symbolic Losses.** In this paper, we have dealt with a specific form of distributional constraint. Conversely, there has been a plethora of work exploring the integration of *hard* symbolic constraints into the learning of neural networks. This can take the form of enforcing a hard constraint [3], whereby the network's predictions are guaranteed to satisfy the pre-specified constraints. Or it can take the form of a soft constraint [48, 34, 1, 4, 2, 5] whereby the network is trained with an additional loss term that penalizes the network for placing any probability mass on predictions that violate the constraint. While in this work we focus on discrete linear inequality constraints defined over binary variables, there is existing work focusing on hybrid linear inequality constraints defined over both discrete and continuous variables and their tractability [10, 55, 54]. The development of inference algorithms for such constraints and their applications such as Bayesian deep learning remain an active topic [52, 28, 53, 51].

## 6 Experiments

In this section, we present a thorough empirical evaluation of our proposed count loss on the three weakly supervised learning problems, *LLP*, *MIL*, and *PU learning*.[1] We refer the readers to the appendix for additional experimental details.

---

[1]Code and experiments are available at `https://github.com/UCLA-StarAI/CountLoss`

Table 4: MIL experiment on the MNIST dataset. Each block represents a different distribution from which we draw bag sizes—First Block: $\mathcal{N}(10, 2)$, Second Block: $\mathcal{N}(50, 10)$, Third Block: $\mathcal{N}(100, 20)$. We run each experiment for 3 runs and report mean test AUC with standard error. The highest metric for each setting is shown in **boldface**.

| Training Bags | 50 | 100 | 150 | 200 | 300 | 400 | 500 |
|---|---|---|---|---|---|---|---|
| Gated Attention | $0.775 \pm 0.034$ | $0.894 \pm 0.012$ | $0.935 \pm 0.005$ | $0.939 \pm 0.006$ | $\mathbf{0.963 \pm 0.002}$ | $0.959 \pm 0.002$ | $\mathbf{0.966 \pm 0.003}$ |
| Attention | $0.807 \pm 0.026$ | $\mathbf{0.913 \pm 0.006}$ | $\mathbf{0.940 \pm 0.004}$ | $0.942 \pm 0.007$ | $0.957 \pm 0.002$ | $0.961 \pm 0.005$ | $0.965 \pm 0.004$ |
| CL (Ours) | $\mathbf{0.818 \pm 0.024}$ | $0.906 \pm 0.009$ | $0.929 \pm 0.005$ | $\mathbf{0.946 \pm 0.001}$ | $0.952 \pm 0.004$ | $\mathbf{0.962 \pm 0.002}$ | $0.963 \pm 0.002$ |
| Gated Attention | $\mathbf{0.943 \pm 0.005}$ | $0.949 \pm 0.009$ | $\mathbf{0.970 \pm 0.005}$ | $\mathbf{0.977 \pm 0.001}$ | $0.983 \pm 0.002$ | $0.986 \pm 0.004$ | $\mathbf{0.987 \pm 0.002}$ |
| Attention | $0.936 \pm 0.010$ | $\mathbf{0.962 \pm 0.006}$ | $\mathbf{0.970 \pm 0.001}$ | $\mathbf{0.977 \pm 0.002}$ | $0.981 \pm 0.002$ | $\mathbf{0.987 \pm 0.001}$ | $\mathbf{0.987 \pm 0.002}$ |
| CL (Ours) | $0.939 \pm 0.010$ | $0.960 \pm 0.002$ | $0.964 \pm 0.007$ | $0.972 \pm 0.002$ | $\mathbf{0.982 \pm 0.003}$ | $0.982 \pm 0.001$ | $\mathbf{0.987 \pm 0.002}$ |
| Gated Attention | $0.975 \pm 0.003$ | $0.981 \pm 0.004$ | $0.992 \pm 0.002$ | $0.987 \pm 0.004$ | $\mathbf{0.996 \pm 0.001}$ | $\mathbf{0.998 \pm 0.001}$ | $0.990 \pm 0.004$ |
| Attention | $\mathbf{0.984 \pm 0.001}$ | $0.982 \pm 0.001$ | $\mathbf{0.996 \pm 0.000}$ | $0.987 \pm 0.007$ | $0.992 \pm 0.004$ | $0.994 \pm 0.002$ | $0.998 \pm 0.000$ |
| CL (Ours) | $0.981 \pm 0.007$ | $\mathbf{0.989 \pm 0.000}$ | $\mathbf{0.996 \pm 0.002}$ | $0.995 \pm 0.001$ | $\mathbf{0.996 \pm 0.002}$ | $0.993 \pm 0.003$ | $\mathbf{0.999 \pm 0.001}$ |

Table 5: MIL: We report mean test accuracy, AUC, F1, precision, and recall averaged over 5 runs with std. error on the Colon Cancer dataset. The highest value for each metric is shown in **boldface**.

| Method | Accuracy | AUC | F1 | Precision | Recall |
|---|---|---|---|---|---|
| Gated Attention | $0.909 \pm 0.014$ | $0.908 \pm 0.013$ | $0.886 \pm 0.021$ | $0.916 \pm 0.020$ | $0.879 \pm 0.020$ |
| Attention | $0.893 \pm 0.015$ | $0.890 \pm 0.008$ | $0.876 \pm 0.017$ | $0.908 \pm 0.016$ | $0.879 \pm 0.018$ |
| CL (Ours) | $\mathbf{0.915 \pm 0.008}$ | $\mathbf{0.912 \pm 0.010}$ | $\mathbf{0.903 \pm 0.010}$ | $\mathbf{0.936 \pm 0.014}$ | $\mathbf{0.898 \pm 0.007}$ |

## 6.1 Learning from Label Proportions

We experiment on two datasets: 1) *Adult* with $8192$ training samples where the task is to predict whether a person makes over $50k$ a year or not given personal information as input; 2) *Magic Gamma Ray Telescope* with $6144$ training samples where the task is to predict whether the electromagnetic shower is caused by primary gammas or not given information from the atmospheric Cherenkov gamma telescope [19].[2]

We follow Scott and Zhang [40] where two settings are considered: one with label proportions uniformly on $[0, \frac{1}{2}]$ and the other uniformly on $[\frac{1}{2}, 1]$. Additionally, we experiment on a third setting with label proportions distributing uniformly on $[0, 1]$ which is not considered in Scott and Zhang [40] but is the most natural setting since the label proportion is not biased toward either 0 or 1. We experiment on four bag sizes $n \in \{8, 32, 128, 512\}$.

Count loss (CL) denotes our proposed approach using the loss objective defined in Table 2 for LLP. We compare our approach with a mutual contamination framework for LLP (LMMCM) [40] and against Proportion Loss (PL) [43].

**Results and Discussions** We show our results in Table 3. Our method showcases superior results against the baselines on both datasets and variations in bag sizes. Especially in cases with lower bag sizes, i.e., 8, 32, CL greatly outperforms all other methodologies. Among our baselines are methods that approximate the bag posterior (PL), which we show to be less effective than optimizing the exact bag posterior with CL.

## 6.2 Multiple Instance Learning

We first experiment on the MNIST dataset [30] and follow the MIL experimental setting in Ilse et al. [24]: the training and test set bags are randomly sampled from the MNIST training and test set respectively; each bag can have images of digits from 0 to 9, and bags with the digit 9 are labeled positive. Moreover, the dataset is constructed in a balanced way such that there is an equal amount of positively and negatively labeled bags as in Ilse et al. [24]. The task is to train a classifier that is able to predict bag labels; the more challenging task is to *discover key instances*, that is, to train a classifier that identifies images of digit 9. Following Ilse et al. [24], we consider three settings that vary in the bag generation process: in each setting, bags have their sizes generated from a normal distribution being $\mathcal{N}(10, 2), \mathcal{N}(50, 10), \mathcal{N}(100, 20)$ respectively. The number of bags in

---

[2]Publicly available at `archive.ics.uci.edu/ml`

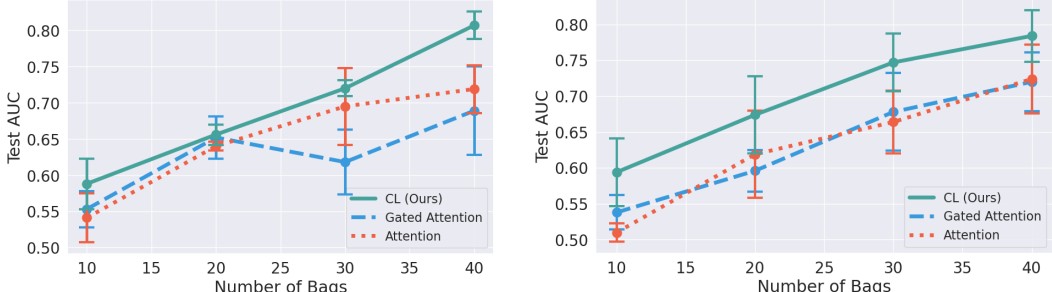

Figure 2: MIL MNIST dataset experiments with decreased numbers of training bags and lower bag size. Left: bag sizes sampled from $\mathcal{N}(10, 2)$; Right: bag sizes sampled from $\mathcal{N}(5, 1)$. We plot the mean test AUC (aggregated over 3 trials) with standard errors for 4 bag sizes. Best viewed in color.

training set $n$ is in $\{50, 100, 150, 200, 300, 400, 500\}$. Thus, we have $3 \times 7 = 21$ settings in total. Additionally, we introduce experimental analysis on *how the performance of the learning methods would degrade as the number of bags and total samples in training set decreases*, by modulating the number of training bags $n$ to be $\{10, 20, 30, 40\}$ and selecting bag sizes from $\mathcal{N}(5, 1)$ and $\mathcal{N}(10, 2)$.

We also experiment on the Colon Cancer dataset [41] to simulate a setting where bag instances are not independent. The dataset consists of 100 total hematoxylin-eosin (H&E) stained images, each of which contains images of cell nuclei that are classified as one of: epithelial, inflammatory, fibroblast, and miscellaneous. Each image represents a bag and instances are $27 \times 27$ patches extracted from the original image. A

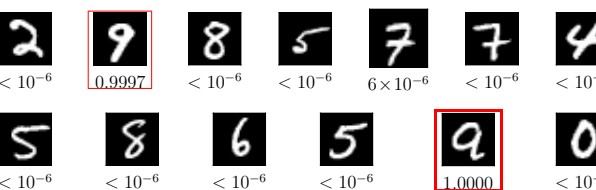

Figure 3: A test bag from our MIL experiments, where we set only the digit 9 as a positive instance. Highlighted in red are digits identified to be positive with corresponding probability beneath.

positively labeled bag or image is one that contains the epithelial nuclei. For both datasets, we include the Attention and Gated Attention mechanism [24] as baselines. We also use the MIL objective defined in Table 2.

**Results and Discussions** For the MNIST experiments, CL is able to outperform all other baselines or exhibit highly comparable performance for bag-level predictions as shown in Table 4. A more interesting setting is to compare how robust the learning methods are if the number of training bags decreases. Wang et al. [44] claim that instance-level classifiers tend to lose against embedding-based methods. However, we show in our experiment that this is not true in all cases as seen in Figure 2. While Attention and Gated Attention are based on embedding, they suffer from a more severe drop in predictive performance than CL when the number of training bags drops from 40 to 10; our method shows great robustness and consistently outperforms all baselines. The rationale we provide is that with a lower number of training instances, we need more supervision over the limited samples we have. Our constraint provides this additional supervision, which accounts for the difference in performance.

We provide an additional investigation in Figure 3 to show that our approach learns effectively and delivers accurate instance-level predictions under bag-level supervision. In Figure 3, we can see that even though the classifier is trained on feedback about whether a bag contains the digit 9 or not, it accurately discovers all images of digit 9. To reinforce this, Table 7 and Table 8, in Appendix B, show that our approach outperforms existing instance-space methods on instance-level classification.

Our experimental results on the Colon Cancer dataset are shown in Table 5. We show that both our proposed objectives are able to consistently outperform baseline methods on all metrics. Interestingly, we do not expect CL to perform well when instances in a bag are dependent; however, the results indicate that our count loss is robust to these settings.

Table 6: PU Learning: We report accuracy and standard deviation on a test set of unlabeled data, which is aggregated over 3 runs. The results from CVIR, nnPU, and uPU are aggregated over 10 epochs, as in Garg et al. [22], while we choose the single best epoch based on validation for our approaches. The highest metric for each setting is shown in **boldface**.

| Dataset | Network | CL-expect (Ours) | CL (Ours) | CVIR | nnPU | nPU |
|---------|---------|------------------|-----------|------|------|-----|
| Binarized MNIST | MLP | $95.9 \pm 0.15$ | $\mathbf{96.4 \pm 0.01}$ | $96.3 \pm 0.07$ | $96.1 \pm 0.14$ | $95.2 \pm 0.19$ |
| MNIST17 | MLP | $98.7 \pm 0.17$ | $\mathbf{99.0 \pm 0.19}$ | $98.7 \pm 0.09$ | $98.4 \pm 0.20$ | $98.4 \pm 0.09$ |
| Binarized CIFAR | ResNet | $79.2 \pm 0.27$ | $80.1 \pm 0.34$ | $\mathbf{82.3 \pm 0.18}$ | $77.2 \pm 1.03$ | $76.7 \pm 0.74$ |
| CIFAR Cat vs. Dog | ResNet | $\mathbf{76.5 \pm 1.86}$ | $74.8 \pm 1.64$ | $73.3 \pm 0.94$ | $71.8 \pm 0.33$ | $68.8 \pm 0.53$ |

## 6.3 Learning from Positive and Unlabeled Data

We experiment on dataset MNIST and CIFAR-10 [29], following the four simulated settings from Garg et al. [22]: 1) Binarized MNIST: the training set consist of images of digits $0-9$ and images with digits in range $[0, 4]$ are positive instances while others as negative; 2) MNIST17: the training set consist of images of digits 1 and 7 and images with digit 1 are defined as positive while 7 as negative; 3) Binarized CIFAR: the training set consists of images from ten classes and images from the first five classes is defined as positive instances while others as negative; 4) CIFAR Cat vs. Dog: the training set consist of images of cats and dogs and images of cats are defined as positive while dogs as negative. The mixture proportion is $0.5$ in all experiments. The performance is evaluated using the accuracy on a test set of unlabeled data.

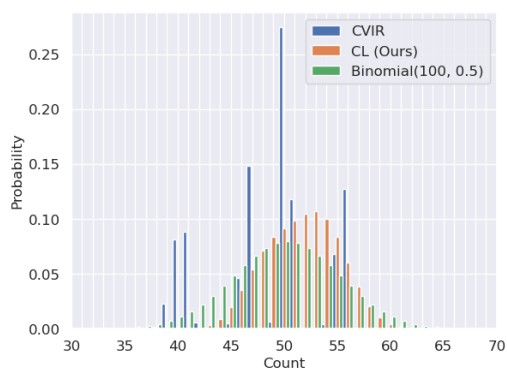

Figure 4: MNIST17 setting for PU Learning: We compute the average discrete distribution for CL and CVIR, over 5 test bags, each of which contain 100 instances. A ground truth binomial distribution of counts is also shown.

As shown in Table 2, we propose two objectives for PU learning. Our first objective is denoted by CL whereas the second approach is denoted by CL-expect. We compare against the Conditional Value Ignoring Risk approach (CVIR) [22], nnPU [26], and uPU [37].

**Results and Discussions** Accuracy results are presented in Table 6 where we can see that our proposed methods perform better than baselines on 3 out of the 4 simulated PU learning settings. CL-expect builds off a similar "exactly-k" count approach, which we have shown to work well in the label proportion setting. The more interesting results are from CL where we fully leverage the information from a distribution as supervision instead of simply using the expectation. We think of this as applying a loss on each count weighted by their probabilities from the binomial distribution. We provide further evidence that our proposed count loss effectively guides the classifier towards predicting a binomial distribution as shown in Figure 4: we plot the count distributions predicted by CL and CVIR as well as the ground-truth binomial distribution. We can see that CL is able to generate the expected distribution, proving the efficacy of our approach.

## 7 Conclusions

In this paper, we present a unified approach to several weakly-supervised tasks, i.e., LLP, MIL, PU. We construct our approach based on the idea of using weak labels to constrain count-based probabilities computed from model outputs. A future direction for our work can be to extend to multi-class classification as well as explore the applicability to other weakly-supervised settings, e.g. label noise learning, semi-supervised learning, and partial label learning [15, 36, 58].

## Acknowledgments

We would like to thank Yuhang Fan for helpful discussions. This work was funded in part by the DARPA PTG Program under award HR00112220005, the DARPA ANSR program under award FA8750-23-2-0004, NSF grants #IIS-1943641, #IIS-1956441, #CCF-1837129, and a gift from RelationalAI. GVdB discloses a financial interest in RelationalAI. ZZ is supported by an Amazon Doctoral Student Fellowship.

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
