## A Proofs

**Lemma A.1.** *Let $R_{llp}$ be our risk estimator defined over $p(\boldsymbol{x}, \tilde{y})$ as $R_{llp}(f) = \frac{1}{k(k+1)} \mathbb{E}_{p(\boldsymbol{x}^k, \tilde{y})}[\ell(f(\boldsymbol{x}), \boldsymbol{y})]$. Following the assumptions in Section 3.1 from Kobayashi et al. [27], our proposed method is risk-consistent.*

*Proof.* In Kobayashi et al. [27], it is shown that the risk $R$ in classical multi-class classification can be reduced to a risk $R_{rc}$ over $p(\boldsymbol{x}^k, \tilde{y}^k)$ as shown in Equation 1 in Kobayashi et al. [27] under certain assumptions.

Consider binary classification and follow our notations, we rewrite the Equation 1 in Kobayashi et al. [27] as below,

$$R_{rc}(f) = \frac{1}{k(k+1)} \mathbb{E}_{p(\boldsymbol{x}^k, \tilde{y})}$$

$$\sum_{\boldsymbol{y} \in \mathcal{Y}^k} \frac{\prod_{j=1}^k p(y_j \mid \boldsymbol{x}_j)}{\sum_{\boldsymbol{y}' \in \mathcal{Y}^k, \sum_j y'_j = \tilde{y}} \prod_{j=1}^k p(y'_j \mid \boldsymbol{x}_j)} \ell(f(\boldsymbol{x}^k), \boldsymbol{y})$$

We notice that the weight term attached to the loss can be further rewritten as a constrained probability as follows,

$$\frac{\prod_{j=1}^k p(y_j \mid \boldsymbol{x}_j)}{\sum_{\boldsymbol{y}' \in \mathcal{Y}^k, \sum_j y'_j = \tilde{y}} \prod_{j=1}^k p(y'_j \mid \boldsymbol{x}_j)} = p(\boldsymbol{y} \mid \sum_{j=1}^k y_j = \tilde{y}, \boldsymbol{x}^k)$$

This allows us to further rewrite the risk $R_{rc}$ with likelihood loss being $\ell(f(\boldsymbol{x}^k), \boldsymbol{y}) = -p(\sum_{j=1}^k y_j = k\tilde{y} \mid \boldsymbol{x}^k)$:

$$R_{rc}(f) = \frac{1}{k(k+1)} \mathbb{E}_{p(\boldsymbol{x}^k, \tilde{y})}$$

$$\left[ -\sum_{\boldsymbol{y} \in \mathcal{Y}^k} p(\boldsymbol{y} \mid \sum_{j=1}^k y_j = k\tilde{y}, \boldsymbol{x}^k) p(\sum_{j=1}^k y_j = k\tilde{y} \mid \boldsymbol{x}^k) \right]$$

$$= \frac{1}{k(k+1)} \mathbb{E}_{p(\boldsymbol{x}^k, \tilde{y})} \left[ -\sum_{\boldsymbol{y} \in \mathcal{Y}^k} p(\boldsymbol{y}, \sum_{j=1}^k y_j = k\tilde{y} \mid \boldsymbol{x}^k) \right]$$

$$= \frac{1}{k(k+1)} \mathbb{E}_{p(\boldsymbol{x}^k, \tilde{y})} \left[ -p(\sum_{j=1}^k y_j = k\tilde{y} \mid \boldsymbol{x}^k) \right]$$

$$= \frac{1}{k(k+1)} \mathbb{E}_{p(\boldsymbol{x}^k, \tilde{y})}[\ell(f(\boldsymbol{x}^k), \boldsymbol{y})] = R_{llp}(f)$$

The last few lines follow from the definition of conditional probabilities. This shows that the risk $R_{rc}(f) = R_{llp}(f)$, meaning that the reduction from risk $R_{rc}(f)$ to the classical risk $R(f)$ in Kobayashi et al. [27] is applicable to our risk estimator $R_{llp}$, which proves that our learning method is risk-consistent. $\square$

**Proposition A.2.** *Assume that the loss function $\ell(f(\boldsymbol{x}), y)$ is $\rho$-Lipschitz with respect to $f(\boldsymbol{x})$ for any $y \in \mathcal{Y}$ bounded by some constant. Let $f_{llp}$ be the hypothesis that minimizes the empirical risk, and $f_{llp}^*$ is the hypothesis that minimizes the true risk, then $f_{llp}$ converges to $f_{llp}^*$ as $m \to \infty$.*

*Proof.* This claim immediately follows Lemma A.1, where we shows that $R_{rc}(f) = R_{llp}(f)$. Therefore, it holds that $R_{llp}(\hat{f}) - R_{llp}(f^*) = R_{(sc)}(\hat{f}) - R_{(sc)}(f^*)$, where the latter term, an always positive term, is shown in Theorem 3.1 in Kobayashi et al. [27] that it converges to $0$ at rate $\sqrt{m}$. $\square$

**Proposition** 4.1 *The count probability $p(\sum_{i=1}^{k} \hat{y}_i = s)$ of sampling $k$ prediction variables with summation being $s$ from an unconstrained distribution $p(\boldsymbol{y}) = \prod_{i=1}^{k} p(\hat{y}_i)$ can be computed exactly in time $\mathcal{O}(ks)$. Moreover, the set $\{p(\sum_{i=1}^{k} \hat{y}_i = s)\}_{s=0}^{k}$ can also be computed in time $\mathcal{O}(k^2)$.*

*Proof.* The claim that $p(\sum_{i=1}^{k} \hat{y}_i = s)$ can be computed exactly in time $\mathcal{O}(ks)$ follows immediately from Proposition 1 in Ahmed et al. [6]: in Ahmed et al. [6], the unconstrained distribution is a factorized distribution obtained from $k$ outputs from a single neural network model while in our case, the unconstrained distribution $p(\boldsymbol{y})$ is obtained from applying a classifier that gives a single output $p(y_i)$ on $k$ inputs; the constructive proof of Proposition 1 in Ahmed et al. [6] still applies in our case. Moreover, the computation of $p(\sum_{i=1}^{k} \hat{y}_i = k)$ is done in a dynamic programming manner in the sense that for any $s < k$, $p(\sum_{i=1}^{k} \hat{y}_i = s)$ is an intermediate result for computing $p(\sum_{i=1}^{k} \hat{y}_i = k)$. By caching the intermediate result, the set $\{p(\sum_{i=1}^{k} \hat{y}_i = s)\}_{s=0}^{k}$ can be obtained by the time $p(\sum_{i=1}^{k} \hat{y}_i = k)$ is computed, which finishes our proof. $\qquad\square$

## B  Instance MIL Experimental Results

In this section, we provide results for instance level feedback in the MIL setting. The baselines that we used in our experiments, Gated-Attention and Attention are both examples of embedding based approaches and do not make instance-level predictions. We compare against one baseline approach, which is based on Instance-Max from Ilse et al. [24]. This uses the maximum instance probability as an approximation for the "positiveness" of a bag. We then train it with a binary cross entropy. Note that max pooling is stated in the literature as the best performing option and makes the "most sense" in the MIL setting [24, 44].

Table 7: MIL experiment on MNIST dataset on instance-level classification. Each block represents a different distribution from which we draw bag sizes—First Block: $\mathcal{N}(10, 2)$, Second Block: $\mathcal{N}(50, 10)$, Third Block: $\mathcal{N}(100, 20)$. We run each experiment for 3 runs and report mean test accuracy with standard error. We bold the highest value and both if the standard-errors overlap.

| Training Bags | 50 | 100 | 150 | 200 | 300 | 400 | 500 |
|---|---|---|---|---|---|---|---|
| Instance-Max | $0.8714 \pm 0.0015$ | $0.9577 \pm 0.0096$ | $0.9494 \pm 0.0232$ | $\mathbf{0.9845 \pm 0.0009}$ | $0.9885 \pm 0.0004$ | $\mathbf{0.9903 \pm 0.0008}$ | $0.9908 \pm 0.0004$ |
| CL (Ours) | $\mathbf{0.9551 \pm 0.0055}$ | $\mathbf{0.9780 \pm 0.0015}$ | $\mathbf{0.9826 \pm 0.0014}$ | $\mathbf{0.9864 \pm 0.0005}$ | $\mathbf{0.9906 \pm 0.0001}$ | $\mathbf{0.9905 \pm 0.0007}$ | $\mathbf{0.9916 \pm 0.0003}$ |
| Instance-Max | $0.9398 \pm 0.0010$ | $0.9415 \pm 0.0008$ | $0.9513 \pm 0.0113$ | $0.9686 \pm 0.0123$ | $\mathbf{0.9849 \pm 0.0010}$ | $0.9848 \pm 0.0008$ | $\mathbf{0.9867 \pm 0.0008}$ |
| CL (Ours) | $\mathbf{0.9732 \pm 0.0009}$ | $\mathbf{0.9776 \pm 0.0009}$ | $\mathbf{0.9799 \pm 0.0010}$ | $\mathbf{0.9816 \pm 0.0005}$ | $\mathbf{0.9839 \pm 0.0013}$ | $\mathbf{0.9864 \pm 0.0006}$ | $\mathbf{0.9865 \pm 0.0014}$ |
| Instance-Max | $0.9446 \pm 0.0007$ | $0.9462 \pm 0.0005$ | $0.9583 \pm 0.0076$ | $0.9700 \pm 0.0035$ | $0.9750 \pm 0.0017$ | $0.9776 \pm 0.0008$ | $0.9695 \pm 0.0097$ |
| CL (Ours) | $\mathbf{0.9695 \pm 0.0010}$ | $\mathbf{0.9717 \pm 0.0011}$ | $\mathbf{0.9759 \pm 0.0013}$ | $\mathbf{0.9764 \pm 0.0006}$ | $\mathbf{0.9780 \pm 0.0001}$ | $\mathbf{0.9805 \pm 0.0008}$ | $\mathbf{0.9798 \pm 0.0003}$ |

Table 8: MIL experiment on MNIST dataset on instance-level classification. Each block represents a different distribution from which we draw bag sizes—First Block: $\mathcal{N}(10, 2)$, Second Block: $\mathcal{N}(50, 10)$, Third Block: $\mathcal{N}(100, 20)$. We run each experiment for 3 runs and report mean test AUC with standard error. We bold the highest value and both if the standard-errors overlap.

| Training Bags | 50 | 100 | 150 | 200 | 300 | 400 | 500 |
|---|---|---|---|---|---|---|---|
| Instance-Max | $0.4904 \pm 0.0054$ | $0.8171 \pm 0.0465$ | $0.7740 \pm 0.1072$ | $0.9288 \pm 0.0064$ | $0.9460 \pm 0.0022$ | $\mathbf{0.9562 \pm 0.0037}$ | $0.9603 \pm 0.0016$ |
| CL (Ours) | $\mathbf{0.8341 \pm 0.0135}$ | $\mathbf{0.9040 \pm 0.0146}$ | $\mathbf{0.9291 \pm 0.0070}$ | $\mathbf{0.9394 \pm 0.0005}$ | $\mathbf{0.9571 \pm 0.0021}$ | $\mathbf{0.9592 \pm 0.0029}$ | $\mathbf{0.9647 \pm 0.0012}$ |
| Instance-Max | $0.4956 \pm 0.0007$ | $0.4965 \pm 0.0003$ | $0.5960 \pm 0.0821$ | $0.7297 \pm 0.0959$ | $0.8566 \pm 0.0088$ | $0.8554 \pm 0.0080$ | $\mathbf{0.8733 \pm 0.0048}$ |
| CL (Ours) | $\mathbf{0.7518 \pm 0.0090}$ | $\mathbf{0.7900 \pm 0.0081}$ | $\mathbf{0.8125 \pm 0.0106}$ | $\mathbf{0.8261 \pm 0.0064}$ | $\mathbf{0.8473 \pm 0.0064}$ | $\mathbf{0.8717 \pm 0.0063}$ | $\mathbf{0.8709 \pm 0.0120}$ |
| Instance-Max | $0.4974 \pm 0.0002$ | $0.5007 \pm 0.0016$ | $0.6170 \pm 0.0571$ | $0.7099 \pm 0.0311$ | $0.7546 \pm 0.0164$ | $0.7792 \pm 0.0080$ | $0.7102 \pm 0.0867$ |
| CL (Ours) | $\mathbf{0.7008 \pm 0.0077}$ | $\mathbf{0.7214 \pm 0.0102}$ | $\mathbf{0.7617 \pm 0.0130}$ | $\mathbf{0.7673 \pm 0.0059}$ | $\mathbf{0.7832 \pm 0.0011}$ | $\mathbf{0.8085 \pm 0.0084}$ | $\mathbf{0.8007 \pm 0.0032}$ |

Our results show that for bags of size less than or equal to $150$, our method greatly improves upon the baseline and is better for bag sizes greater than or equal to $200$. We notice that across both methods, performance goes down as bag size increases; we expect this because we have less supervision on positive bags (at least 1 label is less meaningful for bigger bags). However, our approach is able to recover this gap compared to the baseline methodology. In the case of less overall training bags, less than $150$ training bags, we find that Instance-max really suffers on AUC while our objective guides the model to learning something more meaningful—showcasing the robustness of our methodology.

# C Experimental Details

In this section, we will provide relevant training details as it relates to each of our settings including hyperparameters and dataset details.

Table 9: Illustration of Adult and Magic datasets showing the number of training bags for each bag size. Note that we test on the same number of instances in all variations of bag size for both experiments: 16280 for Adult and 3804 for Magic. The breakdown of training bags is the same across all distributions of label proportion as well, i.e., $[0, \frac{1}{2}], [\frac{1}{2}, 1], [0, 1]$.

| Bag Size | Training Bags Adult | Training Bags Magic |
|----------|---------------------|---------------------|
| 8 | 1024 | 768 |
| 32 | 256 | 192 |
| 128 | 64 | 48 |
| 512 | 16 | 12 |

## C.1 Label Proportion

### C.1.1 Adult Dataset

**Hyperparameters.** We use a learning rate of $0.00001$ with the Adam Optimizer and $\beta_1 = 0.9, \beta_2 = 0.999$. The weight decay value is set to $0.001$. We also notice that adding in $L1$ regularization of $0.001$ improved the performance of our method. We train for $10000$ epochs and use a set number of warm epochs for our experiments. All parameters were obtained by using a holdout of $12.5\%$ of training data for validation on the $[0, 1]$ uniform setting. The network shown in Table 10 was also obtained grid search on this same validation set.

Table 10: Network used for Adult dataset in LLP Experiments.

| Layer | Type |
|-------|------|
| 1 | fc - 2048 + ReLU |
| 2 | fc - 64 + ReLU |
| 3 | fc - 1 + logsigmoid |

**Training Procedure.** For CL, we use the parameters and network described in the previous paragraph and early stopping criterion based on validation loss from a held out validation set ($12.5\%$ of training data). For PL, we use the parameters and network except that we do not use $L1$ as we found this improves performance. We also use an early stopping criterion based on validation loss from a held out validation set ($12.5\%$ of training data).

**Computing Resources.** Trained on Intel(R) Xeon(R) CPU E5-2640 v4 @ 2.40GHzU and AMD EPYC 7313P 16-Core Processor CPU.

### C.1.2 Magic Dataset

**Hyperparameters.** We use a learning rate of $0.0001$ with the Adam Optimizer and $\beta_1 = 0.9, \beta_2 = 0.999$. The weight decay value is set to $0.001$. We also notice that adding in $L1$ regularization of $0.001$ improved the performance of our method. We train for $10000$ epochs and use a set number of warm epochs for our experiments. All parameters were obtained by using a holdout of $12.5\%$ of training data for validation on the $[0, 1]$ uniform setting. The network shown in Table 11 was also obtained grid search on this same validation set.

**Training Procedure.** For CL, we use the parameters and network described in the previous paragraph and early stopping criterion based on validation loss from a held out validation set ($12.5\%$ of training data). For PL, we use the parameters and network except that we do not use $L1$ regularization as we found this improves performance. We also use an early stopping criterion

Table 11: Network used for Magic dataset in LLP Experiments.

| Layer | Type |
|-------|------|
| 1 | fc - 2048 + ReLU |
| 2 | fc - 1 + logsigmoid |

based on validation loss from a held out validation set (12.5% of training data). In Table 3, there are two instances where we reran our method with no validation set, i.e. Magic $[0, \frac{1}{2}]$ and Magic $[\frac{1}{2}, 1]$ because early stopping proved to be unstable with a small amount of validation samples. In these experiments, we only use 87.5% of training data and ran for a fixed number of epochs: 2000. This is because with only one validation bag, we can find ourselves with some instability in the training procedure. Note that PL did not benefit from rerunning with this method.

**Computing Resources.** Trained on Intel(R) Xeon(R) CPU E5-2640 v4 @ 2.40GHzU and AMD EPYC 7313P 16-Core Processor CPU.

## C.2  Multi-Instance Learning

### C.2.1  MNIST-Bags

**Dataset Details.** We experiment on various modulations of training bag size and number of training bags. In the main experiment, we draw bag size from: $\{\mathcal{N}(10, 2), \mathcal{N}(50, 10), \mathcal{N}(100, 20)\}$ and modulate number of training bags from $\{50, 100, 150, 200, 300, 400, 500\}$. In total, this makes 21 different settings. In our follow up experiment where we limit the number of training bags and overall bag size, we draw bag size from: $\{\mathcal{N}(5, 1), \mathcal{N}(10, 2)\}$. For each experiment, we sample 1000 test bags with size coorelating to the normal distribution associated.

**Hyperparameters.** All of our hyperparameters derive from Ilse et al. [24]. This includes using the Adam optimizer with $\beta_1 = 0.9, \beta_2 = 0.999$, a learning rate of 0.0005, weight decay of 0.0001, and max epochs of 200. For the main experiment, we use a validation holdout of 20% to find a class weight for balancing the loss on positive bags versus negative bags. (We omit this step for our limited data experiments.)

Table 12: Network used for all MNIST experiments in MIL settings. Derived from the same network shown in Ilse et al. [24].

| Layer | Type |
|-------|------|
| 1 | conv(5, 1, 0) - 20 + ReLU |
| 2 | maxpool(2, 2) |
| 3 | conv(5, 1, 0) - 50 + ReLU |
| 4 | maxpool(2, 2) |
| 5 | fc-500 + ReLU |
| 6 | fc-1 + logsigmoid |

**Training Procedure.** For CL, we train on all the training data for the maximum number of iterations: 200. We also use all of the hyperparameters described in the last paragraph and Ilse et al. [24]. Because we were unable to reproduce the values in Ilse et al. [24] for the Attention and Gated Attention mechanisms, we reran their experiments with our own implementation. To try and reproduce their results, we follow their optimization procedure. Specifically, we use a holdout of training data (20%) and validation loss + error for early stopping. We found that doing so provided the best values for Attention and Gated Attention.

**Instance Pooling.** To pool together instance level classification at the final stage, there are several operations that have been considered in the literature. Some include using the max and mean operator [44]. We propose a new method based on our constraint. We compute the relevant probabilities defined in 3 for the MIL setting. More specifically, we compute the probability that a bag has at least

one positive instance. We then round the probability of at least one positive instance to obtain our bag level classification.

**Computing Resources.** Trained on AMD EPYC 7313P 16-Core Processor CPU.

### C.2.2 Colon Cancer Dataset

**Dataset Details.** The dataset consists of 100 H&E images of which we use 99 of them. There are a total of 51 positive bags and 48 negative bags. We use a series of data augmentations including flipping, cropping, and rotation[3]. Note that these data augmentations do not align with those in the original paper by Ilse et al. [24], so we reran their baseline methods.

**Hyperparameters.** We derive our set of hyperparameters from Ilse et al. [24]. We use the Adam optimizer for all experiments with $\beta_1 = 0.9, \beta_2 = 0.999$. This includes weight decay of 0.0005, learning rate of 0.0001, and a maximum of 100 epochs.

Table 13: MIL: Network used for CL in colon cancer dataset. Derived from the same network shown in Ilse et al. [24].

| Layer | Type |
|:---:|:---:|
| 1 | conv(4, 1, 0) - 36 + ReLU |
| 2 | maxpool(2, 2) |
| 3 | conv(3, 1, 0) - 48 + ReLU |
| 4 | maxpool(2, 2) |
| 5 | fc-512 + ReLU |
| 6 | dropout |
| 7 | fc - 512 + ReLU |
| 8 | dropout |
| 9 | fc-2 + logsigmoid |

**Training Procedure.** We perform 10-fold cross-validation and average the mean value of each metric over 5 seeds. For CL, we do not use early stopping and train on all data for the maximum number of epochs using the hyperparameters mentioned in the previous paragraph. For our baselines, Attention and Gated-Attention, we use the same hyperparameters as mentioned above. However, we follow the optimization procedure detailed in Ilse et al. [24] to give try and reproduce the results given in the paper. This involves using a held out validation set for early stopping with validation loss + error as the stopping criteria. For this experiment, this validation set is assumed to be the size of 1 fold or one-ninth of the training data. (We find that including early stopping helps increase performance for both baselines.)

**Computing Resources.** Trained on NVIDIA RTX A6000 GPU.

### C.3 PU Learning

### C.3.1 MNIST Dataset

**Dataset Details.** Our settings derive from Garg et al. [22]. We construct two main datasets from the original MNIST dataset. This includes the Binarized MNIST and MNIST-17 as detailed in Table 15. In the Binarized MNIST setting, we assign digits $[0-4]$ as positive and $[5-9]$ as negative. In the MNIST-17 setting, we assign digit 1 as positive and 7 as negative. The test set for both settings are chosen from a set of unlabeled data.

**Hyperparameters.** We fix weight decay to be 0.0005 and Adam optimizer for all experiments with $\beta_1 = 0.9, \beta_2 = 0.999$. We use a learning rate of 0.0001 and train for a maximum of 2000 epochs in all experiments for both CL and CL-expect. We use a validation set with size equal to 10% of training data in order to weigh the loss on positive data versus loss on unlabeled data.

---

[3]Refer to https://github.com/utayao/Atten_Deep_MIL for the preprocessed data generation code

Table 14: Network used for MNIST data in PU Learning experiments. Resembles the network in Garg et al. [22] except we replace the last layer with a single output and logsigmoid instead of softmax.

| Layer | Type |
|-------|------|
| 1 | fc - 5000 + ReLU |
| 2 | fc - 5000 + ReLU |
| 3 | fc - 50 + ReLU |
| 4 | fc-1 + logsigmoid |

**Training Procedure.** For MNIST dataset experiments, we use a fully connected multi-layer perceptron (MLP) defined in Table 14. We train CL and CL-expect with the hyperparameters defined in the previous paragraph. Furthermore, we use a held out validation set, equivalent to $10\%$ of training data, for early stopping. While as results in Garg et al. [22] are aggregated over 10 epochs, we choose to pick a single epoch based on our early stopping as this makes the most sense for our optimization technique.

**Computing Resources.** Trained on a singular NVIDIA RTX 2080-Ti GPU.

Table 15: Table taken almost directly from Garg et al. [22]. Table shows the break down of the various simulated PU datasets that we train on.

| Dataset | Simulated PU Dataset | P vs N | Training Positive | Training Unlabeled | Test Unlabeled |
|---------|---------------------|--------|----------|-----------|-----------|
| CIFAR | Binarized CIFAR | $[0-4]$ vs. $[5-9]$ | 12500 | 12500 | 2500 |
| | CIFAR Cat vs. Dog | 3 vs. 5 | 3000 | 3000 | 500 |
| MNIST | Binarized MNIST | $[0-4]$ vs. $[5-9]$ | 15000 | 15000 | 2500 |
| | MNIST-17 | 1 vs. 7 | 3000 | 3000 | 500 |

### C.3.2 CIFAR Dataset.

**Dataset Details.** Our settings derive from Garg et al. [22]. We construct two main datasets from the original CIFAR dataset. This includes the Binarized CIFAR and CIFAR Cat vs. Dog as detailed in Table 15. In the Binarized CIFAR setting, we assign classes $[0-4]$ as positive and classes $[5-9]$ as negative. In the CIFAR Cat vs. Dog setting, we assign Cats (class 3) as positive and Dogs (class 5) as negative. The test set for both settings are chosen from a set of unlabeled data.

**Hyperparameters.** We fix weight decay to be 0.0005 and Adam optimizer for all experiments with $\beta_1 = 0.9, \beta_2 = 0.999$. We use a learning rate of 0.0001 for all experiments except for CL-expect in the CIFAR Cat vs. Dog setting where we use 0.001. We use a validation set with size equal to $10\%$ of training data in order to weigh the loss on positive data versus loss on unlabeled data.

**Training Procedure.** We use a ResNet-18 architecture for all CIFAR experiments. We train CL and CL-expect with the hyperparameters defined in the previous paragraph. Furthermore, we use a held out validation set, equivalent to $10\%$ of training data, for early stopping. While as results in Garg et al. [22] are aggregated over 10 epochs, we choose to pick a single epoch as this makes the most sense for our optimization technique.

**Computing Resources.** Trained on a singular NVIDIA 2080-Ti GPU.

### C.3.3 Early Stopping

The early stopping procedure that we used in our experiments was a bit unique. Using our holdout of validation data, we do early stopping using the proximity to the class prior and validation loss to break ties. We can imagine that if we perfectly identify all positive and unlabeled samples and then calculate accuracy against the actually provided labels, we would get an accuracy equivalent to the class prior. This is because all the positive samples in the unlabeled set would be labeled incorrect.

# D  Limitations

In MIL, one assumption that our approach makes is that the label distribution of instances within a bag are independent. This is a common assumption in the literature as stated in Carbonneau et al. [13]. However, in most practical scenarios, this assumption does not hold. Through empirical validation we show that while this is true, our method can still outperform state of the art benchmarks on the Colon Cancer dataest [41], which violates the independence assumption (Table 5). In PU Learning, our approaches—CL and CL-expect—assume that the batch of data is sufficiently large such that the distribution is roughly binomial. Note that CL-expect relies on the expected value of the binomial distribution while as CL relies on the entire distribution. If the batch of data was too small, this assumption would certainly degrade as batch-to-batch variance would make our loss unsuitable.

# E  Broader Impact

We provide a unified framework to count-based weakly supervised learning. One benefit enjoyed by our approach is that it does not necessitate the presence of instance labels, and is therefore privacy preserving. In many of our settings, we show that we can still train a strong instance level classifier even with these weak bag-level annotations. Another important use case that we have not fully explored in this paper is debiasing classifiers through using our proportion loss as a regularization term. If we know the expected class priors, we can penalize any bias in the classifiers predictions. Care should be taken however since, just as our approach can be used to de-bias classifiers, it can also be used by a malicious actor to bias them.