# OpenReview forum: "A Unified Approach to Count-Based Weakly Supervised Learning"
_NeurIPS.cc/2023/Conference — NeurIPS 2023 poster_

### Official Review · Reviewer_5jbX · 2023-06-29

**Soundness:** 3 good
**Presentation:** 3 good
**Contribution:** 2 fair
**Rating:** 5
**Confidence:** 4

**Summary:**

The paper describes a unified loss for different weakly supervised paradigms: learning from label proportions (LLP), multiple instance learning (MIL) and learning from positive and unlabeled data (PU).
The main idea is to compute the probability of a count sum constraint by summing over all possible combination of events compatible with the constraint (e.g.: exactly k out of n outputs being set to true). An efficient dynamic programming algorithm is given to compute the distribution of the count. It is demonstrated empirically that the count loss can be used to train models in the mentioned weakly supervised settings.

**Strengths:**

The unification of different weakly supervised paradigms under the count loss is a novel contribution. The dynamic programming algorithm provided make the efficient computation of the loss possible.
The paper is easy to follow. The three weakly supervised problems addressed by the paper in details (and potentially more, where the loss can be applied) are well researched and have many real world application.


**Weaknesses:**

In its current form the loss is defined only for binary classification problems, while alternatives like probabilistic logic based solutions are applicable to multi class settings as well.
For example we know that on an image (our bag) there are 5 object, 2 cats, 2 dogs, and 1 apple. In defense of the paper however, It does not seem too complicated  to extend to this case.

The novelty of the loss can be questioned. Probabilistic logic can be used to formulate this and more general losses, and this is actually used in deep learning see [1,2]. For example [1] describes the supervision of MNIST digit bags with their sum, this is clearly more general, as in binary case will give back the count loss. Naively evaluating the probabilistic program computing this probability is of course inefficient, however there are techniques (tabled execuition, memoization) to reuse results of sub-problems quite similarly to dynamic programming. I am not sure if your dynamical programming method exactly corresponds to any of the methods used in probabilistic logic programming, but claiming superiority in computational efficiency would require comparison.

Note that I do not question the novelty of the unified view on LLP, MIL and PU.

[1] Manhaeve, Robin, Sebastijan Dumancic, Angelika Kimmig, Thomas Demeester, and Luc De Raedt. "Deepproblog: Neural probabilistic logic programming." Advances in neural information processing systems 31 (2018).

[2] Oldenhof, Martijn, Adam Arany, Yves Moreau, and Edward De Brouwer. "Weakly Supervised Knowledge Transfer with Probabilistic Logical Reasoning for Object Detection." In The Eleventh International Conference on Learning Representations. 2023. (**NOTE:** this is contemporaneous with your submission, and as such you are not required to be aware of it, **I am ignoring the content of this specific paper when evaluating novelty.**)



**Questions:**

You defined the problem for binary classification,  how difficult would it be to generalize to multiple classes? I assume the main question is the applicability of the algorithm for this case.

How much time the loss computation and the backpropagation through the dynamics programming graph takes proportionally in the experiments. Is it significantly slower to compute than the traditional losses you compare with?

Please do not only bold numerically equal mean values as with 3 repeats there is no difference for example between 0.818 ± 0.024 and 0.807 ± 0.026 (in Table 4).  Check the significance of the results, but at least as a shortcut method bold multiple results if they are so close that their variance significantly overlaps not only if their mean precisely equal. This would considerably help the job of the reviewer and the reader.

Count probability is sometimes computed exactly with probabilistic logic reasoning systems. Would be quite valuable to compare runtime with a probabilistic logic engine using tabled execution. I would execute simply a few case with high $n$ to see computational times.
I also suggest to mention the connection with logical inference in the paper. This gives for example a "possible world" semantic for the sum in the objective, containing the indicator variable.

NOTE: After the authors rebuttal: the clarification of the contribution. and the discussion on the logic programming solution, I raised the score by one.

**Limitations:**

The author acknowledges some limitations, like that it is not currently working for multi-class cases, but there are no detailed discussion on limitations.

---

> ### Author Rebuttal · Authors · 2023-08-09
>
> We thank reviewer 5jbX for such detailed insights. We are pleased to hear that you found the paper easy to follow and the unification as a strong novelty.
>
> **[Multi-class Extension Possibility]**
>
> The multi-class extension is certainly possible though admittedly, it comes with an exponential time trade off with respect to the number of classes. We posted pseudocode for a non vectorized, unbatched version for clarity as a proof of concept in response to reviewer jVGG.
>
> **[Loss Computation and Back Propagation Complexity + Timing]**
>
> “How much time the loss computation and the backpropagation through the dynamics programming graph takes proportionally in the experiments. Is it significantly slower to compute than the traditional losses you compare with?”
>
> | Bag Size  | 8                   | 32                  | 128                 | 512                 |
> |-----------|---------------------|---------------------|---------------------|---------------------|
> | CL (Ours) | $0.1139 \pm 0.0102$ | $0.1270 \pm 0.0155$ | $0.1574 \pm 0.0179$ | $0.4076 \pm 0.0209$ |
> | PL        | $0.1032 \pm 0.0119$ | $0.0981 \pm 0.0042$ | $0.1101 \pm 0.0726$ | $0.1069 \pm 0.0073$ |
>
> Note that the table's units are in seconds.
>
> We present empirical timing results for the LLP setting (Adult with $k \in [0, \frac{1}{2}]$) against a baseline PL (proportion loss) [4]. We time over 10 epochs (getting rid of the first) and then average it over three runs. We report mean and standard deviation We observe that for bag sizes less than or equal to 128, our timing is very comparable. We know that our algorithm’s computation scales with bag size, which accounts for the performance of 512. Hence, we demonstrate that in actual training scenarios, with moderate bag sizes, our algorithm can be trained at comparable efficiency to current approaches.
>
> **[Result Highlighting]**
>
> “Please do not only bold numerically equal mean values as with 3 repeats there is no difference for example between 0.818 ± 0.024 and 0.807 ± 0.026 (in Table 4). Check the significance of the results, but at least as a shortcut method bold multiple results if they are so close that their variance significantly overlaps not only if their mean precisely equal. This would considerably help the job of the reviewer and the reader.”
>
> Thank you for the suggestion. We will include this in our camera-ready submission. Note that the results we presented for instance-level classification in our rebuttal pdf Table $1$ and Table $2$ does this.
>
> **[Probabilistic Logic Engine and DeepProbLog]**
>
> “The novelty of the loss can be questioned. Probabilistic logic can be used to formulate this and more general losses, and this is actually used in deep learning see [1,2]. For example [1] describes the supervision of MNIST digit bags with their sum, this is clearly more general, as in binary case will give back the count loss. Naively evaluating the probabilistic program computing this probability is of course inefficient, however there are techniques (tabled execuition, memoization) to reuse results of sub-problems quite similarly to dynamic programming. I am not sure if your dynamical programming method exactly corresponds to any of the methods used in probabilistic logic programming, but claiming superiority in computational efficiency would require comparison.”
>
> We agree that knowledge compilers, such as the SDD library underlying deepproblog, can handle arbitrary constraints specified in propositional logic. Such compilers take in as input a logical sentence in propositional logic, and output a target form that enables the tractable computation of certain probabilistic queries e.g. the probability of the constraint. Note, however, that such target forms might themselves be intractable—they are worst case exponential in the treewidth of the input logical formula, as is the case with MNIST addition, which does not scale to much more than 2 digits.
>
> [1] show the target form for the exactly-k constraint is tractable. In fact, the algorithm presented therein, and echoed in our paper, corresponds to a structured d-DNNF with a right-linear vtree with a vectorized complexity of $O(n)$, which can be further improved to a vectorized complexity of $O(\log{(n)} \log{(k)})$.
>
> In this paper our contribution is framing a class of weakly-supervised learning problems in terms of the simple yet principled problem of satisfying some count loss; showing that not only can we compute the probability of the constraint tractably, but also symmetric functions thereof e.g.  $\geq k$ or at $\leq k$; and providing the empirical evidence for the merits of the approach. It is our view that, our approach subsuming the MNIST addition example, in retrospect, should count as a positive and not a negative.
>
> **[Limitations]**
>
> We refer the reviewer to Appendix C for a discussion on more limitations.
>
> **[References]**
>
> [1] Kareem Ahmed, Zhe Zeng, Mathias Niepert, and Guy Van den Broeck. Simple: A gradient estimator for k-subset sampling. In Proceedings of the International Conference on Learning Representations (ICLR), may 2023.
>
> [2] Manhaeve, Robin, Sebastijan Dumancic, Angelika Kimmig, Thomas Demeester, and Luc De Raedt. "Deepproblog: Neural probabilistic logic programming." Advances in neural information processing systems 31 (2018).
>
> [3] Xu, Jingyi, et al. "A semantic loss function for deep learning with symbolic knowledge." International conference on machine learning. PMLR, 2018.
>
> [4] Kuen-Han Tsai and Hsuan-Tien Lin. Learning from label proportions with consistency regularization. In Asian Conference on Machine Learning, pages 513–528. PMLR, 2020.

---

> > ### Comment · Reviewer_5jbX · 2023-08-11
> > **Reply to Rebuttal**
> >
> > I would like to thank the authors for the work they put in to answering my (and other reviewers') questions in details.
> >
> > The code example provided for the 3 class case is helpful to see possible generalization of the method. It is appreciated.
> > I am also happy to see that computation time is quite manageable.
> >
> > **On issues still open:**
> > I agree with **Q9rM** in that, based on the wording of the paper it is indeed not clear that the DP algorithm is exactly taken from Ahmend et al. This should be made more explicit in the main text.
> > The algorithm is a textbook example of Dynamic Programming, therefore I am not particularly worried of its originality, but clarity should be provided for the reader. As I mentioned in my original review, I was concentrating on the unification part for identifying novelty, not the algorithm itself.
> >
> > The unification is, according to best of my knowledge, indeed novel. However, note that the formulation in LLP and MIL is quite simple to derive (as also noted by **jVGG**), The KL-divergence based PU loss is more of an original idea, but than it is not clear that it outperforms the expectation matching variant.
> >
> > **My follow-up question:**
> > As you mention SSD and similar libraries "output a target form that enables the **tractable** computation of **certain** probabilistic queries" Can SDD come up automatically with an efficient solution for the counting problem in binary case? (Not the MNIST sum what I mentioned before as an example)

---

> > > ### Author Response · Authors · 2023-08-12
> > >
> > > Thank you for your reply.
> > >
> > > **[algorithm attribution]**
> > >
> > > We're happy to emphasize the attribution of the algorithm to Ahmed et al. in the paper. We would like to emphasize, however, that Ahmed et. al do not consider functions of the exactly-k constraints e.g. at-least $k$ or at-most $k$.
> > >
> > > **[Simplicity]**
> > >
> > > We would like to emphasize that **simplicity is a key strength of our paper**. It is our view that **many papers are published with convoluted techniques that apply to a singular setting**. The fact that we can do so well with such a *simple* approach is an enormous contribution of our work.
> > >
> > > **[Knowledge Compilation]**
> > >
> > > In practice, the compilation process represents a major bottleneck. To see this, consider that one way to write the exactly-$k$ constraint is by enumerating all $n$ choose $k$ subsets and disjoining them. This is clearly not doable for even modestly large values of $n$ and $k$. Other encodings of the exactly-$k$ constraint might fare slightly better, but are still tricky to compile: all such compilers are bottom-down, so the order in which you parse the logical formula, and consequently grow and restrict the function, is crucial. A much better alternative is to follow the algorithm provided to *construct the circuit*. Explicitly creating and maintaining the graph is of little use to us, however, in this case. More details can be seen below.
> > >
> > > We hope this convinces the reviewer to change their recommendation towards an accept, and are happy to answer any more questions.
> > >
> > > **[details]**
> > >
> > > Start at the root of the circuit with an OR node that has two AND nodes as children. Each of the AND nodes in turn has two children, $X1$ and a circuit representing the Boolean function exactly($n-1$, $k-1$), and $\lnot X1$ and a circuit representing the Boolean function exactly($n-1$, $k$). Repeat recursively. This construct a computational graph, in a similar manner to what PyTorch does to perform auto-diff.

---

> > > > ### Comment · Reviewer_5jbX · 2023-08-14
> > > > **Raised score by one point.**
> > > >
> > > > Thank you for clarification. It seems to me that indeed it is not trivial to achieve optimization results similar to here automatically from a logical program, even if it is not entirely sure that it is not possible. I appreciate the work of the authors, and the replies provided to my questions. I raised the score by one.

---

### Official Review · Reviewer_YHmN · 2023-07-01

**Soundness:** 3 good
**Presentation:** 3 good
**Contribution:** 3 good
**Rating:** 7
**Confidence:** 4

**Summary:**

This paper proposes a unified framework for weakly-supervised binary classificasion, where instances can be considered as bags and their weak supervision can be seen as count of positive instances in each bag.
The framework is based on the probability of the count and the authors derived loss functions based on it for three different weakly-supervised settings, Learning from label proportions (LLP), multiple instance learning (MLP), and Positive and unlabeled learning (PU). The direct handling count is reasonable and the information of bag size is incorporated as supervision in contrast to Proportion Loss (PL).
Experimental results on multiple datasets demonstrated that the proposed method performs better than existing methods.

**Strengths:**

- The proposed unified approach for count-based weakly-supervised learning is novel, and the direct handling count in their formulation is reasonable. The solution is exact, not approximation.

- The paper is well-written and easy to follow.

- Experimental results on multiple datasets demonstrated the effectiveness of the proposed method.

**Weaknesses:**

### Originality:
- It is better to discuss concrete descriptions of how the existing work trains models.

### Clarity:
- In p.2, for the definition of a set of training data, n should be defined explicitly.
- Table 1 should be Figure 1.
- Table 1-a should be described in the main text.
- In eq at l.4 from the bottom in p.5, there is a redundant bracket.
- On p.8, it is better to clarify how to generate discrete bag sizes from the normal distribution.
- Figures are better to place on top.
- Positions of proposed methods are inconsistent among tables.

**Questions:**

Please see above.

**Limitations:**

This work focuses on binary classification, and enhancement for multi-class cases is discussed as their future work.

---

> ### Author Rebuttal · Authors · 2023-08-09
>
> We thank reviewer YHmN for the review of our paper. We are delighted to hear that you found our solution well-written, easy to follow, and the empirical evaluation effective.
>
> **[Originality]**
>
> “It is better to discuss concrete descriptions of how the existing work trains models.”
>
> We will make sure to have this for the camera-ready, specifically paying more attention to how the baselines we use train models. Many of the existing approaches seek to approximate what we can compute exactly.
> * Proportion Loss (PL) uses the average of instance probabilities (approximation for our computation) in a bag and then applies binary cross entropy against the ground truth proportion [1].
> * We note that the baselines we use for MIL, Gated Attention and Attention, these are embedding space methods, which means that they lack instance level predictions [2]. Their output is a single probability, which is then penalized via binary cross entropy against the MIL label. Other instance-space approaches are noted in [2, 3] and included using the max, mean, and log-sum-exp as pooling operations to approximate the probability of “at least one positive instance”, which is then trained with binary cross entropy. Note again that these instance methods merely approximate our exact computation.
> * In PU Learning, some of our baselines include uuPU and nPU, which derive from first principles methods on how to minimize estimated empirical risk for the PU Learning task using the risk defined on the “positive risk” on positive data and “negative risk” on unlabeled data [4, 5]. They then use a series of loss functions to test their approach and optimize via gradient descent. We derive a loss directly from unlabeled data without any sort of estimation. Furthermore, the view of unlabeled as a bag annotated with the mixture proportion is very different from prior approaches.
>
> **[Clarity]**
>
> Thank you for the notes on clarity, we will make the proper revisions in lieu of the camera-ready.
>
> **[References]**
>
> [1] Kuen-Han Tsai and Hsuan-Tien Lin. Learning from label proportions with consistency regularization. In Asian Conference on Machine Learning, pages 513–528. PMLR, 2020.
>
> [2] Ilse, Maximilian, Jakub Tomczak, and Max Welling. "Attention-based deep multiple instance learning." International conference on machine learning. PMLR, 2018.
>
> [3] Wang, Xinggang, et al. "Revisiting multiple instance neural networks." Pattern Recognition 74 (2018): 15-24.
>
> [4] Ryuichi Kiryo, Gang Niu, Marthinus C Du Plessis, and Masashi Sugiyama. Positive-unlabeled learning with non-negative risk estimator Advances in neural information processing systems, 30, 2017. 10
>
> [5] M. C. du Plessis, G. Niu, and M. Sugiyama. Analysis of learning from positive and unlabeled data. In NIPS, 2014.

---

### Official Review · Reviewer_rn5b · 2023-07-03

**Soundness:** 3 good
**Presentation:** 3 good
**Contribution:** 3 good
**Rating:** 6
**Confidence:** 3

**Summary:**

The paper proposes a unified approach for three problems of weak supervision: LLP, MIL, and PU learning. The approach is based on estimation of a common computational block – count probability. For its estimation, the dynamic programming method is used. The method is compared in three problem scenarios with different baselines.

**Strengths:**

Developing accurate models for learning with weak supervision (LLP, MIL) is a challenging and important problem. Different weak supervision scenarios share commonalities in their problem formulation, and this paper clearly outlines the common part, and the main bottleneck — the count probability. Having count probability estimated, one can plug it into the scenario-dependent objective.

Overall, the paper is written very clearly and easy to follow, provides a good overview of weak supervision scenarios and unifies them under the same umbrella of optimization problems.


**Weaknesses:**

The main part I missed in the paper is the training algorithm. The paper describes in detail the inference part of it: given the estimations of instance-probabilities $p(y_i|x_i)$ it shows how one can compute the count probability. But what I missed is how this is used in the optimization part for training, how are the gradients propagated from the objective to the instance level? It might be straightforward, but in my opinion including the pseudocode/explanation of the training method is essential to understand the details. Without that, I failed to understand the training method, and therefore the value of the paper.

----
Raised my score after reading the rebuttal.

**Questions:**

My main question is how the training algorithm actually works, and can the authors provide a pseudocode for that?

**Limitations:**

It would be great to see the runtime depending on the bag size, in other words is there any computational limitation to the approach related to the parameters of the problem?

---

> ### Author Rebuttal · Authors · 2023-08-09
>
> We thank reviewer rn5b for the comments. We are happy that you found the paper well-written and easy to follow.
>
> **[Training Algorithm]**
>
> “My main question is how the training algorithm actually works, and can the authors provide a pseudocode for that?”
>
> Let us consider the example of LLP. As noted by the reviewer, our classifier produces $p(\hat{y}_i = 1 |x_i)$, which for the sake of brevity we will call $p(\hat{y}_i = 1)$. Given that we have a bag of $n$ samples and that our true count of positive samples is $k$, we want to compute: $p(\sum_i \hat{y}_i = k)$ using $p(\hat{y}_i = 1)$ for all integers $i \in [[1, n]]$. This is done via the “exactly-k” algorithm. Our goal is to *maximize this probability*. Hence, we minimize the objective: $-\log{p(\sum_i \hat{y}_i = k)}$. Note that this objective is certainly *differentiable* as the DP algorithm, shown in Algorithm 1, reduces to products, sums, exponentials, and logarithms and hence, we can *backpropagate* through it.
>
> Pseudocode with pytorch for LLP training:
>
> ```
> # labels are the ground truth count of each bag in the batch
> # prob_pred_equals_k is oracle call to our dp algorithm, returning a log probability
> def train():
>    for epoch in range(epochs):
>       for train_batch, labels in training_generator:
>          optimizer.zero_grad()
>          instance_predictions = model(train_batch)
>
>          batch_loss = -1*prob_pred_equals_k(instance_predictions, labels, bag_size)
>          loss = batch_loss.mean()
>
>          loss.backward()
>          optimizer.step()
> ```
>
> **[Runtime Limitation]**
>
> “It would be great to see the runtime depending on the bag size, in other words is there any computational limitation to the approach related to the parameters of the problem?”
>
> The runtime of our algorithm is vectorized $O(n)$, where $n$ is the bag size, which means that the computation linearly scales with bag size. We present some empirical results in the following table, which is for the LLP setting with the Adult dataset where $k \in [0, \frac{1}{2}]$.
>
> | Bag Size  |          8          |          32         |         128         |         512         |
> |-----------|:-------------------:|:-------------------:|:-------------------:|:-------------------:|
> | CL (Ours) | $0.1139 \pm 0.0102$ | $0.1270 \pm 0.0155$ | $0.1574 \pm 0.0179$ | $0.4076 \pm 0.0209$ |
> | PL        | $0.1032 \pm 0.0119$ | $0.0981 \pm 0.0042$ | $0.1101 \pm 0.0726$ | $0.1069 \pm 0.0073$ |
>
> Our baseline is proportion loss [1], which uses an approximation (mean of instance probabilities) for the probability that we compute exactly. We time over 10 epochs (getting rid of the first) and then average it over three runs. We report mean and standard deviation in seconds. We see that for bag sizes less than or equal to 128, our timing is comparable but degrades in the case of a very large bag, i.e. 512.
>
> [1] Kuen-Han Tsai and Hsuan-Tien Lin. Learning from label proportions with consistency regularization. In Asian Conference on Machine Learning, pages 513–528. PMLR, 2020.

---

> > ### Comment · Reviewer_rn5b · 2023-08-13
> >
> > Thanks for the reply! I see now how the loss can be backpropagated. I will raise my score.

---

### Official Review · Reviewer_Q9rM · 2023-07-07

**Soundness:** 4 excellent
**Presentation:** 4 excellent
**Contribution:** 4 excellent
**Rating:** 7
**Confidence:** 3

**Summary:**

The author proposes a differentiable count loss for three weak supervision settings: LLP, MIL, PU learning. The paper provides a clear explanation of the proposed loss, demonstrates strong empirical performance, and highlights the efficiency of its implementation using dynamic programming.

**Strengths:**

1. The paper provides a clear explanation of the proposed differentiable count loss, making it easier for readers to understand the methodology and its application in weak supervision settings.

2. The paper demonstrates strong empirical performance across various tasks and datasets. This provides evidence of the effectiveness of the proposed loss and enhances the credibility of the research.

3. The authors showcase the efficiency of their implementation by utilizing dynamic programming. This highlights the practicality and computational advantages of the proposed approach.

**Weaknesses:**

1. The paper could benefit from a clearer articulation of its contribution. Providing a more explicit comparison and emphasizing the novelty of the proposed method would strengthen the paper; The author mentioned that prior works (Ardehaly&Culotta and Tsi&Lin) are motivated by the same principle. More comparison to the prior work would be helpful.

2. The paper lacks sufficient citations to related works; in the related work section “there exists some literature …, for example …” Expanding the reference list and acknowledging existing literature, would enhance the paper's academic rigor.

**Questions:**

1. "We also perform risk analysis with details in Appendix?" Could you specify which appendix contains the details of the risk analysis?

2.  Bottom of page 5: There appears to be a typo in the expression "p(s_1 ≤ ∑_i y_i) ≤ s_2)". Please correct this typo.

3. In Proposition 4.1, it is unclear whether the dynamic programming approach is proposed or if it is directly using the result from Ahmed[1]. Please clarify this point.

**Limitations:**

The authors have adequately addressed the limitations.

---

> ### Author Rebuttal · Authors · 2023-08-09
>
> We would like to thank reviewer Q9rM for the feedback. We are glad you found approach's efficiency, explanation, and applicability to be solid.
>
> **[Articulating Contribution]**
>
> “The paper could benefit from a clearer articulation of its contribution. Providing a more explicit comparison and emphasizing the novelty of the proposed method would strengthen the paper”
>
> Our contribution is a unifying framework for count-based weakly supervised settings. We use DP approach given by [1] and are the first ones to extend the loss function to train models. We also apply functions of "exactly-k" in our MIL and PU Learning approach. We point the reviewer to the general rebuttal section, where we articulated our contributions and novelty in more detail.
>
> **[Existing/Prior Work]**
>
> “More comparison to the prior work would be helpful.”
>
> Thank you for this note. We will make sure to update our manuscript with more existing and prior work than we already have. We elaborated on prior "unifying" methods for weakly supervised learning, other weakly supervised learning paradigms, and also previous approaches that approximate our findings. (More detail in the general rebuttal.)
>
> **[Typos/Clarification]**
>
> “We also perform risk analysis with details in Appendix?" Could you specify which appendix contains the details of the risk analysis?”
>
> Appendix A
>
> Thank you for pointing out the other typos, they have been updated in a new version of the manuscript for the camera-ready.
>
> **[DP Approach]**
>
> “In Proposition 4.1, it is unclear whether the dynamic programming approach is proposed or if it is directly using the result from Ahmed[1]. Please clarify this point.”
>
> The DP approach for computing “exactly-k” was indeed first proposed by [1] but we would like to reiterate our contributions. We extend the evaluation to functions of "exactly-k" as in the MIL setting (at least $1$ computation) and PU setting (penalizing the count distribution against a binomial via KL Divergence). Furthermore, we are the first to use it as an actual loss function.
>
> **[References]**
>
> [1] Kareem Ahmed, Zhe Zeng, Mathias Niepert, and Guy Van den Broeck. Simple: A gradient estimator for k-subset sampling. In Proceedings of the International Conference on Learning Representations (ICLR), may 2023.

---

### Official Review · Reviewer_jVGG · 2023-07-27

**Soundness:** 3 good
**Presentation:** 3 good
**Contribution:** 2 fair
**Rating:** 5
**Confidence:** 4

**Summary:**

This paper proposes a unification of three different kinds of weak supervision, viz., Learning from label proportions (LLP), Multi-instance learning (MIL), and learning from positive and unlabeled data (PU learning). They all involve supervision at the bag level instead of the instance level (except for PU learning, where some positive instances are also given).
The unification comes from the fact that in all three cases, we need to compute the probability distribution of a sum of binary variables using the probability distribution over each binary variable. To do so efficiently, the authors use the dynamic-programming algorithm proposed by Ahmed et al. 23. Once the distribution over sum is computed, it can easily be used to define the likelihood-based loss function for each of the three scenarios. For example, for LLP, one would directly maximize $log Pr(\sum_{j \in B_i} y_j = k_i)$ for a given $(B_i, k_i)$ train sample, and efficient computation of $log Pr(\sum_{j \in B_i} y_j = k_i)$ is done using DP algorithm proposed by Ahmed et al. 23


**Strengths:**

The paper is very well written, and the technique is clearly explained with all the assumptions and proofs.

I like the idea of computing the exact probabilities for the sum of binary variables and using it to define the loss function for the three scenarios of weak supervision considered in the paper.




**Weaknesses:**

1. The crux of the paper lies in computing the probability over the sum of binary variables, which has been borrowed from Ahmed et al., 23. Extending the idea to apply it to the scenario of LLP, MIL, and PU is natural and not that novel.

2. The related works section should also cover other paradigms of weak supervision, e.g. partial label learning.



**Questions:**

1. The paper considers the scenario of only binary classification under weak supervision, whereas the evaluation is done against a more generic method (Tsai et al., 20). Is it possible to extend count loss to multi-class classification? Maybe, by considering it as a binary classification task with an additional one-hot constraint that can be enforced using the semantic loss function defined in Xu et al.,17, or other similar techniques.

2. In the MIL experiment, observing the instance-level accuracy obtained by your method will be interesting.

[Xu et al 17]: A Semantic Loss Function for Deep Learning with Symbolic Knowledge


**Limitations:**

The authors have sufficiently addressed the limitations of their work and also discuss the positive impact of weak supervision on the privacy of the subjects whose data is collected for training.

---

> ### Author Rebuttal · Authors · 2023-08-09
>
> We would like to thank reviewer jVGG for the feedback. We are glad they found the paper well-written and enjoyed our approach.
>
> **[Related Work - Partial Label Learning]**
>
> Thank you for the comment, we have included this as part of our general rebuttal section.
>
> **[Limited Novelty]**
>
> We unify $3$ count-based weakly supervised frameworks under one exact, tractable, and differentiable computation and are the first ones to use this as a training objective. Furthermore, we extend the evaluation to *functions* of "exactly-k" as in the MIL setting (at least $k$) and PU setting (penalizing the count distribution against a binomial via KL Divergence). We point the reviewer to the general rebuttal section for more detail of our restatement of our contributions and novelty.
>
> **[Multi-class Classification]**
>
> Yes, our method is certainly extendable to multi-class classification.
>
> Suppose we have a length n bag with label k. For binary classification we optimize the following probability:
> $P(\text{Class}_1 = k, \text{Class}_2 = n - k) = P(\text{Class}_1 = k)$
>
> Now suppose we still have a bag length $n$. For multi-class, assume $\text{Class}_1$ has count $a_1$, $\text{Class}_2$ with count $a_2$, $\text{Class}_3$ with count $a_3$, we want to optimize:
> $P(\text{Class}_1 = a_1, \text{Class}_2 = a_2, \text{Class}_3 = a_3) = P(\text{Class}_1 = a_1, \text{Class}_2 = a_2)$
> , where $a_1 + a_2 + a_3 = n$. (Note that if we fix $a_1, a_2$, then $a_3$ is automatically fixed.)
>
> Consider the following non vectorized, non batched dynamic algorithm for $3$ classes. Note that this can easily extend to more than $3$ classes.
>
> ```
> def prob_equals_k_3(probs, class_1_count, class_2_count, bag_size):
>    dp = torch.zeros(bag_size + 1, class_1_count + 1, class_2_count + 1)
>    dp[0,0,0] = 1
>    for k in range(1, bag_size + 1):
>       for i in range(0, class_1_count + 1):
>          for j in range(0, class_2_count + 1):
>             if i + j > k:
>                continue
>             if i >= 1:
>                dp[k, i, j] += dp[k - 1, i - 1, j]*probs[k - 1, 0]
>             if j >= 1:
>                dp[k, i, j] += dp[k - 1, i, j - 1]*probs[k - 1, 1]
>             dp[k,i,j] += dp[k - 1][i][j]*(1 - probs[k - 1, 0] - probs[k - 1, 1])
>    return dp[bag_size, class_1_count, class_2_count]
> ```
>
> In the above algorithm, probs has shape (bag_size, $2$). Furthermore, $\text{probs}[k - 1][0]$ is the probability that the kth instance is class 1, $\text{probs}[k - 1][1]$ is the probability that the kth instance is class 2, and (1 - $\text{probs}[k - 1][0]$ - $\text{probs}[k - 1][1]$) is the probability that the kth instance is class 3 (note the indexing subtlety here). $dp[k, i, j]$ represents the probability that among the first $k$ instances, $i$ of them are class $1$, $j$ of them are class $2$, and $k - i - j$ of them are class $3$.
>
> **[Instance Level Performance]**
>
> We show our results in the general rebuttal pdf in Table $1$ and Table $2$.
>
> We take our baseline from [1], which is called Instance-max. For a bag of instances, Instance-max uses the maximum predicted probability as an approximation for probability that there exists at least one positive instance. We then train it with a binary cross entropy. Note that max pooling is stated in the literature as the best performing option and makes the “most sense” in the MIL setting [1, 2].
>
> Our results show that for bags of size less than or equal to $150$, our method greatly improves upon the baseline and is better for bag sizes greater than or equal to $200$. We notice that across both methods, performance goes down as bag size increases; we expect this because we have less supervision on positive bags (at least $1$ label is less meaningful for bigger bags). However, our approach is able to recover this gap compared to the baseline methodology. In the case of *less* overall training bags, i.e. $\leq 150$, we find that Instance-max really suffers on AUC while our objective guides the model to learning something more meaningful—showcasing the robustness of our methodology.
>
> **[References]**
>
> [1] Ilse, Maximilian, Jakub Tomczak, and Max Welling. "Attention-based deep multiple instance learning." International conference on machine learning. PMLR, 2018.
>
> [2] Wang, Xinggang, et al. "Revisiting multiple instance neural networks." Pattern Recognition 74 (2018): 15-24.

---

> > ### Comment · Reviewer_jVGG · 2023-08-15
> >
> > Dear authors
> >
> > Thanks for a thoughtful rebuttal and for addressing the concerns raised. The pseudo-code for 3-class classification is helpful, and the additional experiment comparing instance-level accuracy in MIL is appreciated.
> >
> > I agree with reviewer 5jbX that the unification of the three scenarios is indeed novel, though straightforward. In my opinion, the **backpropagable** DP algorithm to compute the probability of the sum is the key and since that has been borrowed from Ahmed et. al., I am not sure if this paper crosses the bar for NeurIPS. Nevertheless, I am raising my score by 1.
> >
> > Regards,

---

### Author Rebuttal · Authors · 2023-08-09

Thank you all for your reviews and insights!

**[Novelty]**

While we do acknowledge that [3] showcases the “exactly k” dynamic programming algorithm we employ, our novelty lies in the *unified view* of weakly supervised learning for *count-based* settings. We are the first to leverage the dynamic programming algorithm as a loss function to train models end to end. We also extend the applicability of this loss to *functions* of “exactly-k”. This is seen in the construction of the MIL objective function, which computes the probability of at least $1$ positive instance in a bag. Furthermore, in our PU learning approach, we penalize the KL Divergence between the count distribution produced by the dynamic programming algorithm to the ground truth binomial distribution.

**[Related Work]**

One concern brought up by many of the reviewers was the Related Works section.

We will add a segment on additional weakly supervised learning settings, incorporating descriptions and citations for the following settings:

**Weakly Supervised Settings:**
* Semi-Supervised Learning: Semi-supervised learning is a close relative to PU Learning with the difference being that labeled samples can be both positive and negative [7].
* Label Noise Learning: Label-noise learning derives from a common real-life scenario: mislabeled instances [11]. There are many variations but two common ones involve whether label noise is independent or dependent on instances [2].
* Partial Label Learning: In partial label learning, each instance is provided a set of labels of which exactly one is true [1].

(Note that we believe that some of these settings could potentially utilize our “exactly-k” approach but this is a task for future work to explore.)

We will also add the following to the **Count Loss** section:
In MIL settings, among instance level classifiers, there are also several ways of approximating the bag posterior. These include mean, max, and log-sum-exp pooling to approximate the likelihood that a bag has *at least one positive instance* [4]. But again, these are all *approximations* of what our "at least $1$" computation does exactly. In PU Learning, to our best knowledge, the view of unlabeled data as a bag annotated with the mixture proportion is novel and yet to be considered.

In the **Unified Approaches** section, we will add information on other approaches:
Co-Training and Self-Training are examples of similar techniques that are applicable to a wide variety of weakly supervised settings [5, 6]. Self-Training involves progressively incorporating more unlabeled data via our model’s prediction (with pseudo-label) and then training a model on more data as an iterative algorithm [11]. Co-Training leverages two models that have different “views” of the data and iteratively augment each other's training set with samples they deem as “well-classified”. They are both traditionally applied to semi-supervised learning tasks but can also extend to multiple instance learning settings [9, 10, 12].

**[Multi-Class Classification]**

Multi-Class classification is certainly possible with our methodology. However, the computation scales exponentially with the number of  classes. This is something we hope to explore in future work.

**[Instance-Level Classification MIL]**

We provide compelling instance-level classification results as part of our rebuttal pdf in Table 1 and Table 2.

**[References]**

[1] Timothee Cour, Ben Sapp, and Ben Taskar. 2011. Learning from Partial Labels. J. Mach. Learn. Res. 12, null (2/1/2011), 1501–1536.

[2] Song, Hwanjun, et al. "Learning from noisy labels with deep neural networks: A survey." IEEE Transactions on Neural Networks and Learning Systems (2022).

[3] Kareem Ahmed, Zhe Zeng, Mathias Niepert, and Guy Van den Broeck. Simple: A gradient estimator for k-subset sampling. In Proceedings of the International Conference on Learning Representations (ICLR), may 2023.

[4] Wang, Xinggang, et al. "Revisiting multiple instance neural networks." Pattern Recognition 74 (2018): 15-24.

[5] Avrim Blum and Tom Mitchell. 1998. Combining labeled and unlabeled data with co-training. In Proceedings of the eleventh annual conference on Computational learning theory (COLT' 98). Association for Computing Machinery, New York, NY, USA, 92–100. https://doi.org/10.1145/279943.279962

[6] David Yarowsky. 1995. Unsupervised word sense disambiguation rivaling supervised methods. In Proceedings of the 33rd annual meeting on Association for Computational Linguistics (ACL '95). Association for Computational Linguistics, USA, 189–196. https://doi.org/10.3115/981658.981684

[7] X. Zhu and A.B. Goldberg. Introduction to semi-supervised learning. Synthesis Lectures on Artificial Intelligence and Machine Learning, 3(1):1–130, 2009.

[8] Xie, Ming-Kun, and Sheng-Jun Huang. "Partial multi-label learning." Proceedings of the AAAI conference on artificial intelligence. Vol. 32. No. 1. 2018.

[9] H. Lu, Q. Zhou, D. Wang and R. Xiang, "A co-training framework for visual tracking with multiple instance learning," 2011 IEEE International Conference on Automatic Face & Gesture Recognition (FG), Santa Barbara, CA, USA, 2011, pp. 539-544, doi: 10.1109/FG.2011.5771455.

[10] Sun, S. A survey of multi-view machine learning. Neural Comput & Applic 23, 2031–2038 (2013). https://doi.org/10.1007/s00521-013-1362-6

[11] Karamanolakis, Giannis, et al. "Self-training with weak supervision." arXiv preprint arXiv:2104.05514 (2021).

[12] Liu, Kangning, et al. "Multiple instance learning via iterative self-paced supervised contrastive learning." Proceedings of the IEEE/CVF Conference on Computer Vision and Pattern Recognition. 2023.

---

### Decision · Program_Chairs · 2023-09-21

**Decision:**

Accept (poster)

**Comment:**

This paper proposes a unified framework for learning in three weakly supervised settings: learning from label proportions, multiple instance learning, ad positive-unlabeled learning. The paper refers to these settings as count-based learning because the maximum likelihood learning in all these cases can be defined in terms of the probability of exactly k out of n random variables being true. The paper shows this computation is differentiable and can be used to efficiently express learning across the three settings. Experiments show that this unified approach is competitive with state of the art methods developed specifically for each of these settings.

The reviewers saw benefits of having a unified approach to these problems that have been studied separately. They also found the experimental results convincing that the approach works well. During the discussion phase, the reviewers made several suggestions that the authors are strongly encouraged to take into account for the final version. First, the reviewers provided additional citations and connections to the areas of related work. Discussing them will strengthen the paper. Second, multiple reviewers found the details of the training algorithms for the various methods unclear. These should be clarified in the final version.